# The chloroplast 2-cysteine peroxiredoxin functions as thioredoxin oxidase in redox regulation of chloroplast metabolism

Mohamad-Javad Vaseghi, Kamel Chibani, Wilena Telman, Michael Florian Liebthal, Melanie Gerken, Helena Schnitzer, Sara Mareike Mueller, Karl-Josef Dietz*

Department of Biochemistry and Physiology of Plants, Faculty of Biology, University of Bielefeld, Bielefeld, Germany

**Abstract** Thiol-dependent redox regulation controls central processes in plant cells including photosynthesis. Thioredoxins reductively activate, for example, Calvin-Benson cycle enzymes. However, the mechanism of oxidative inactivation is unknown despite its importance for efficient regulation. Here, the abundant 2-cysteine peroxiredoxin (2-CysPrx), but not its site-directed variants, mediates rapid inactivation of reductively activated fructose-1,6-bisphosphatase and NADPH-dependent malate dehydrogenase (MDH) in the presence of the proper thioredoxins. Deactivation of phosphoribulokinase (PRK) and MDH was compromised in *2cysprxAB* mutant plants upon light/dark transition compared to wildtype. The decisive role of 2-CysPrx in regulating photosynthesis was evident from reoxidation kinetics of ferredoxin upon darkening of intact leaves since its half time decreased 3.5-times in *2cysprxAB*. The disadvantage of inefficient deactivation turned into an advantage in fluctuating light. Physiological parameters like MDH and PRK inactivation, photosynthetic kinetics and response to fluctuating light fully recovered in *2cysprxAB* mutants complemented with 2-CysPrxA underlining the significance of 2-CysPrx. The results show that the 2-CysPrx serves as electron sink in the thiol network important to oxidize reductively activated proteins and represents the missing link in the reversal of thioredoxin-dependent regulation.
DOI: https://doi.org/10.7554/eLife.38194.001

*For correspondence:
karl-josef.dietz@uni-bielefeld.de

Competing interests: The authors declare that no competing interests exist.

## Introduction

Redox homeostasis is a fundamental property of life and intimately linked to the redox state of glutathione and protein thiols. Deviations from normal thiol redox environment with glutathione redox potentials of about −315 mV elicit alterations in gene expression, posttranscriptional processes, metabolism and also compensatory mechanisms for readjustment of the cellular redox norm (*Foyer and Noctor, 2009*). All cells maintain a thiol redox regulatory network consisting of redox input elements, redox transmitters, redox targets, redox sensors and reactive oxygen species as final electron acceptors (*Dietz, 2008*). The chloroplast thiol network is presumably the best studied case of thiol redox regulation since thiol redox regulation of Calvin-Benson cycle (CBC) enzymes by a thiol factor was discovered as early as in 1971 (*Buchanan et al., 1971*). Later on the decisive factor was proven to be thioredoxin (Trx) (*Holmgren et al., 1977*). Genome annotations and biochemical studies elucidated the complex composition of the plastid thiol network of plastids and of plants in general (*Meyer et al., 2009*; *Buchanan, 2016*).

The plastids of *Arabidopsis thaliana* contain a set of 10 canonical Trxs (Trx-f1, -f2, -m1, -m2, -m3, -m4, -x, -y1, -y2, -z) and additional Trx-like proteins, for example the chloroplast drought-induced stress protein of 32 kDa (CDSP32) (*Broin and Rey, 2003*), four ACHT proteins (*Dangoor et al.,*

2009), the Lilium proteins and Trx-like proteins (*Chibani et al., 2009*; *Meyer et al., 2009*). The canonical Trxs are reduced by ferredoxin (Fd)-dependent thioredoxin reductase (FTR) and themselves reduce oxidized target proteins. The FTR-pathway reduces the Trx-isoforms with distinct efficiency as recently shown by *Yoshida and Hisabori (2017)*. Well-characterized Trx-targets are the CBC enzymes fructose-1,6-bisphosphatase (FBPase), NADPH-dependent glyceraldehyde-3-phosphate dehydrogenase, seduheptulose-1,7-bisphosphatase, ribulose-5-phosphate kinase (PRK) and ribulose-1,5-bisphosphate carboxylase oxygenase activase (RubisCO activase) (*Michelet et al., 2013*). The chloroplast FBPase is reduced by Trx-f with high preference (*Collin et al., 2003*). Another reductively activated target is the NADPH-dependent malate dehydrogenase (MDH) which plays a role in the export of excess reducing equivalents in photosynthesizing chloroplasts. MDH is activated if the stromal reduction potential increases (*Scheibe and Beck, 1979*) under conditions of limited availability of electron acceptors, for example in high light, low temperature or low $CO_2$ (*Hebbelmann et al., 2012*). Activation is mediated by m-type Trxs (*Collin et al., 2003*).

Hundreds of Trx-targets and polypeptides undergoing thiol modifications have been identified in proteome studies (*Montrichard et al., 2009*). The various redox proteomics approaches employed affinity chromatography, differential gel separation and isotope coded-affinity or fluorescence-based labeling (*Mock and Dietz, 2016*). Trapping chromatography using Trx variants with mutated resolving cysteines allowed for efficient identification of Trx-targets (*Motohashi et al., 2009*). The target proteins are essentially associated with all important metabolic activities and molecular processes such as transcription, translation, turnover, defense against reactive oxygen species and also signaling pathways in the chloroplast (*Buchanan, 2016*). The enzymes are often activated upon reduction, but redox regulation of for example signaling components and certain enzymes involves oxidation as part of the response, for example in transcriptional regulation (*Dietz, 2014*; *Giesguth et al., 2015*; *Gütle et al., 2017*). The significance of controlled oxidation is most apparent if considering the metabolic transition from light-driven photosynthesis to darkness or from high to low photosynthetic active radiation. Enzymes of the CBC must be switched off upon darkening or adjusted to the new activity level in decreased light in order to prevent depletion of metabolites and de-energization of the cell (*Gütle et al., 2017*). In fact upon tenfold lowering the irradiance from for example 250 to 25 µmol quanta $m^{-2}$ $s^{-1}$, the $CO_2$ assimilation transiently drops to $CO_2$ release prior to adjustment to the new lower level. The NADPH/NADP$^+$-ratio falls from 1.1 to 0.1 prior to readjustment of the previous ratio of about one in the lower light. Since also the ATP/ADP-ratio drops within 30 s, and thus the assimilatory power, *Prinsley et al. (1986)* concluded, that the deactivation of the enzymes occurs with slight delay, but then enables recovery of appropriate metabolite pools to proceed with carbon assimilation in the new light condition.

Reversible and rapid redox regulation requires efficient mechanisms not only for reduction but also for oxidation of target proteins. The reductive pathway via Trxs is well established, thus the open question concerns the mechanism of oxidation. The likely candidate is hydrogen peroxide as most stable and thus diffusible reactive oxygen species (ROS). The light reactions generate superoxide in the Mehler reaction, and possibly at low rates also in the reaction catalyzed by the plastid terminal oxidase (*Dietz et al., 2016*). Two molecules of superoxide are dismutated to one molecule of $H_2O_2$ and one molecule of $O_2$. In particular, photosynthesis-derived ROS were often considered as unavoidable side products but are now accepted drivers in the redox regulatory network and thus in regulation and signaling (*Driever and Baker, 2011*). ROS likely are involved in operating the regulatory thiol switches (*Groitl and Jakob, 2014*). Direct non-specific reaction of $H_2O_2$ with target proteins would lack specificity. It is also questionable that thiols of target proteins including Trxs as transmitters can compete as electron donors to $H_2O_2$ with those proteins that evolved for that purpose, the thiol peroxidases of the chloroplast (*Dietz, 2016*; *Flohé, 2016*). $O_2$ appears even less suitable than $H_2O_2$ as thiol oxidant due to its relatively high stability.

Thiol peroxidases are evolutionary ancient proteins that efficiently react with peroxides. The catalytic cysteinyl residue is embedded in a specific molecular environment that lowers its p$K_a$ value. The high affinity with $K_M$-values in the low micromolar range compensates for low turnover numbers. The Arabidopsis chloroplasts contains four peroxiredoxins (two 2-cysteine peroxiredoxins [2-CysPrxA, 2-CysPrxB], peroxiredoxin Q [PrxQ], peroxiredoxin IIE [PrxIIE]) and two glutathione peroxidase-like proteins which also function as Trx-dependent peroxidases (*Horling et al., 2003*; *Navrot et al., 2006*; *Dietz, 2016*). The 2-CysPrxAB represent the most abundant chloroplast peroxiredoxin with about 100 µM concentration (*Peltier et al., 2006*). Antisense plants lacking the 2-CysPrx reveal

disturbed photosynthesis and altered redox homeostasis (*Baier and Dietz, 1999*; *Baier et al., 2000*). Analysis of *A. thaliana* lines with T-DNA insertions established that the NADP-dependent thioredoxin reductase C (NTRC) is the predominant and efficient electron donor to 2-CysPrx (*Pulido et al., 2010*) and that the 2-CysPrx participates in detoxification of $H_2O_2$ generated in the Mehler reaction (*Awad et al., 2015*). But it is also established that 2-CysPrx accepts electrons with lower efficiency from various Trx and Trx-like proteins (*Collin et al., 2003*). Recently it was shown that the severe growth phenotype of *ntrc*-plants can be recovered by crossing them with the *2cysprxAB* plants (*Pérez-Ruiz et al., 2017*). In the *ntrc* plants, the FBPase was insufficiently reductively activated in the light. In gel redox analysis revealed that Trx-f remained partially oxidized even in the light and thus probably was unable to activate FBPase. This effect was mostly reverted in the *ntrc/2cysprxAB*-triple mutants. The authors proposed that the lack of NTRC and the concomitant oxidation of 2-CysPrxAB (*Pulido et al., 2010*; *Pérez-Ruiz et al., 2017*) oxidize Trx-f. This assumption is tentatively in line with the low electron donation capacity of Trx-f to 2-CysPrx reported by *Collin et al. (2003)*.

This study aimed to dissect the hypothesis proposed in 2008 by Dietz, namely that the 2-CysPrx functions as the missing link in the thiol-disulfide redox regulatory network, as Trx-oxidase in redox regulation of chloroplast metabolism and other regulatory processes. *In vitro* experiments were designed to study the inactivation of FBPase and MDH by oxidized 2-CysPrx. The inactivation depended on the presence of Trxs with preference for target-specific Trxs. The hypothesis was further scrutinized *in vivo* by comparing wildtype (WT) and *2cysprxAB* for Kautzki effect, ferredoxin reoxidation kinetics, inactivation of malate dehydrogenase and ribulose-5-phosphate kinase upon darkening. The data strongly support the Trx-oxidase hypothesis and show a pre-activation of metabolism in darkness and a delayed inactivation during light-to-dark transfer. Development of a kinetic model allowed for simulating the *in vitro* observations. Finally, it will be shown that the growth inhibition phenotype of *2cysprxAB* relative to WT-plants is reversed in short fluctuating light pulses supporting the hypothesis that 2-CysPrx is needed for efficient inactivation in fluctuating light. While WT fails to use the light energy during the short light pulses, the *2cysprxAB* mutant exploits this energy given its inefficient deactivation of the redox regulated enzymes. Thus 2-CysPrx is a principle component in rapid plant light acclimation.

## Results

*A. thaliana* lacking 2-CysPrx develop with delay and show defects in photosynthesis (*Baier and Dietz, 1999*). *Figure 1* depicts chlorophyll a-fluorescence transients of WT seedlings and seedlings lacking cyclophilin 20–3, an interactor of 2-CysPrx, or *2cysprxAB* when grown on solidified Murashige Skoog plus/minus sucrose. The actinic light was switched on after 16.5 s and both wildtype and *cyp20-3* seedlings displayed the Kautsky peak of chlorophyll-a fluorescence emission, followed by a slow decline to the new steady state. In a contrasting manner, *2cysprxAB* seedlings showed a small fluorescence increase in the absence of sucrose, which was missing in the presence of sucrose. The Kautsky effect reflects the reduction of the electron transport chain in the first phase of illumination followed by reoxidation in the course of activation of NADPH- and ATP-consuming metabolic pathways, in particular the CBC and the malate valve. The lack of the fluorescence peak in *2cysprxAB* was tentatively interpreted as indication that the energy sinks and energy dissipation mechanisms such as non-photochemical or photochemical quenching activities remained active in darkness and that this pre-activation is linked to properties of 2-CysPrx.

Chloroplast fructose-1,6-bisphosphatase (FBPase) and NADPH-dependent malate dehydrogenase (MDH) were selected as two well-known examples for light-activated enzymes in order to address the hypothesis that 2-CysPrx affects the activation state of redox proteins.

Stroma protein was obtained from isolated intact chloroplasts, and chloroplast Trxs and 2-CysPrxA were generated as recombinant protein. The stroma extract was first preincubated in 1 mM dithiothreitol (DTT) for reductive activation of FBPase. The *in vitro* assay after 1:1 dilution (residual DTT concentration of 500 µM) was initiated by addition of the substrate FBP (*Figure 2*). The stromal FBPase efficiently converted FBP to Fru-6-P as seen from the time-dependent increase in absorption. Addition of Trx-f1 slightly increased the activity of FBPase indicating that activation had not been fully achieved during the preincubation, so that Trx-f1 in the presence of the residual 500 µM DTT was able to further reduce and activate the stromal FBPase. The activity of FBPase was unchanged

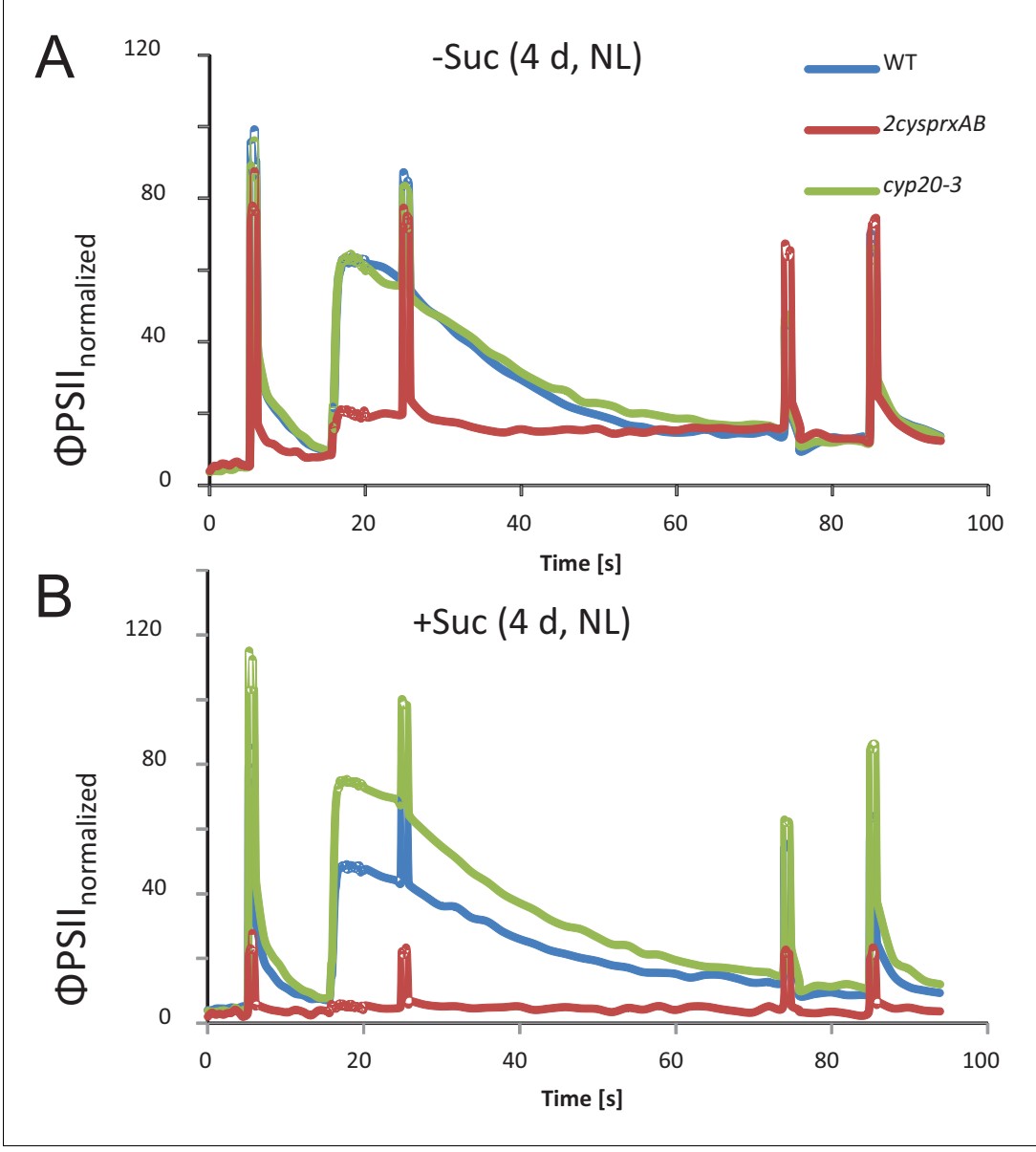

**Figure 1.** Chlorophyll-a fluorescence kinetics of 4d old wildtype, *2cysprxAB* and *cyclophilin 20–3* mutants. Seeds were placed on phytogel-solidified half strength Murashige-Skoog medium with or without 0.5% sucrose, stratified for 2 d and then grown in a growth chamber with 8 hr light, 16 hr dark at 80 µmol quanta $m^{-2}$ $s^{-1}$ and 23°C. Chlorophyll a-fluorescence transients of the seedlings on the plates were imaged with the FluoroCam using the following settings: 30 min dark acclimation, measuring light on after 5 s, actinic light with 65 µmol quanta $m^{-2}$ $s^{-1}$ after 16.5 s. Actinic light was switched off after 76.5 s and the recording terminated at 94.5 s. Saturating light pulses of 900 µmol quanta $m^{-2}$ $s^{-1}$ were applied after 5, 25, 75 and 85 s. Data are means of kinetic data obtained from 34 (*2cysprxAB*) and 69 – 77 seedlings in three independent experiments.

DOI: https://doi.org/10.7554/eLife.38194.002

The following figure supplement is available for figure 1:

**Figure supplement 1.** Chlorophyll-fluorescence kinetics of 7 d old wildtype, *2cysprxAB* and *cyclophilin 20–3* mutants.

DOI: https://doi.org/10.7554/eLife.38194.003

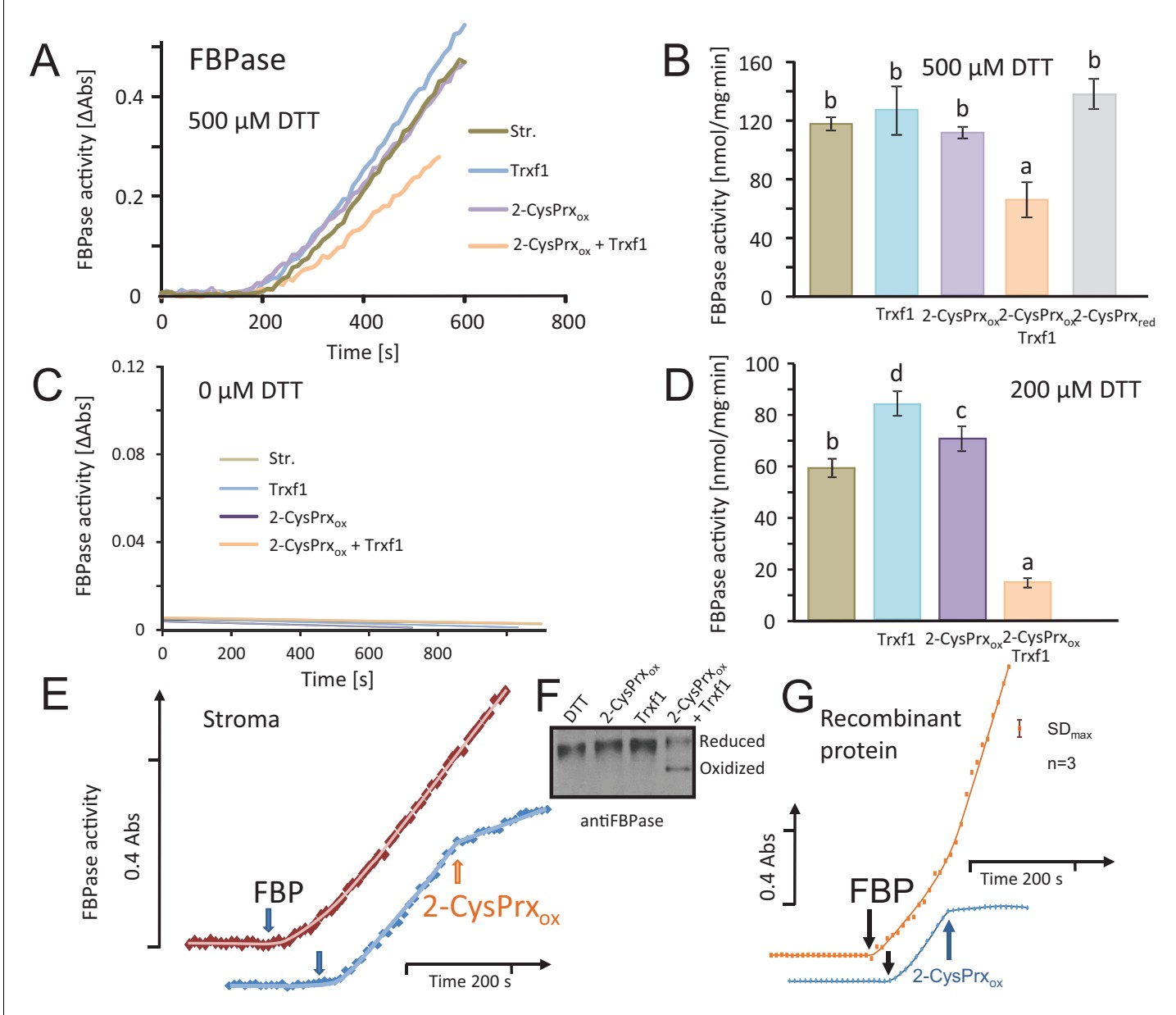

**Figure 2.** Inhibition of reductively activated FBPase by oxidized 2-CysPrxA. (**A**) Isolated stroma was treated with 1 mM dithiothreitol (DTT) plus/minus Trx for pre-activation and then added to the FBPase activity test with a final DTT concentration of 500 μM. Test compounds as indicated were added at t = 0 s. Concentrations were: 5 μM Trx-f1, 5 μM oxidized 2-CysPrx. The enzyme assay was started by addition of 600 μM FBP at t = 180 s. (**B**) Average activity values were calculated from the maximum slope obtained for the different treatments from independent experiments. Data are means ± SD of n = 3. Statistics was calculated using one-sided ANOVA followed by Tukey's HSD. Different letters mark significantly different groups. (**C**) Control measurements without DTT. (**D**) Activity in the presence of final DTT concentration of 200 μM (pre-activation with 400 μM DTT). (**E**) The speed of inactivation was documented by addition of 5 μM 2-CysPrxA$_{ox}$ in the presence of Trx-f1 at the time point as indicated after establishing the linear rate of FBP hydrolysis. The FBPase activity was rapidly inhibited after addition of 2-CysPrxA$_{ox}$. (**F**) Redox state of FBPase in the assay shown in *Figure 2A* after labeling the reduced thiols with mPEG$_{24}$. Migration of the reduced 2-CysPrx was slowed down and was abundant in the samples treated with DTT, DTT plus 2-CysPrx$_{ox}$ or DTT plus Trx-f1 (upper band), but was strongly decreased after addition of the combination of 2-CysPrx$_{ox}$ and Trx-f1. Same results were seen in several experiments. (**G**) Inhibition of recombinant FBPase by oxidized 2-CysPrxA. In this test, following reductive pre-activation of purified recombinant FBPase (2 μM) in the presence of Trx-f1 (5 μM), FBP hydrolysis was initiated by addition of 600 μM FBP. At the time point as indicated, oxidized recombinant 2-CysPrxA at a final concentration of 5 μM was added and caused rapid inhibition of FBPase activity, while the F6P-release continued in the control without 2-CysPrxA. The assay was performed three times with similar result. The error bar (top right) gives the maximum SD observed.

DOI: https://doi.org/10.7554/eLife.38194.004

*Figure 2 continued*

The following source data and figure supplement are available for figure 2:

**Source data 1.** Values from the FBPase activity tests shown in *Figure 2B and D*.
DOI: https://doi.org/10.7554/eLife.38194.006
**Figure supplement 1.** Thioredoxin-specificity of inactivation of FBPase by oxidized 2-CysPrx.
DOI: https://doi.org/10.7554/eLife.38194.005

upon addition of oxidized 2-CysPrx in the absence of Trx. However, if the complete test was reconstituted with reduced stroma, Trx-f1 and oxidized 2-CysPrx, FBPase was significantly inhibited. Apparently, Trx-f1 mediated the oxidative inactivation of FBPase by transferring electrons from reduced FBPase to 2-CysPrx$_{ox}$. As an additional control reduced 2-CysPrx was added to reduced stroma and established the same activity as with Trx-f1 alone (*Figure 2A,B*).

Activity was not seen in the absence of DTT (*Figure 2C*). Overall activation was lower if the pre-activation was performed with 400 µM instead of 1 mM DTT (*Figure 2D*). Under these conditions, addition of reduced Trx-f1 further activated the FBPase. Addition of 2-CysPrx$_{ox}$ to reduced stroma again had no effect on FBPase activity, but the combination of reduced stroma, Trx-f1 and 2-CysPrx$_{ox}$ led to 76% inhibition. The inhibition was rapidly achieved as revealed when 2-CysPrx$_{ox}$ was added to the ongoing enzyme reaction (*Figure 2E*). In parallel, the reduced form of FBPase which runs as upper band in *Figure 2F* due to incorporation of 2 molecules of mPEG$_{24}$ per molecule FBPase was diminished in the fully reconstituted assay with reduced stroma, Trx-f1 and 2-CysPrx$_{ox}$. The same inhibitory effect was seen in an assay where exclusively recombinant proteins were used (*Figure 2G*).

The efficiency of inactivation depended on Trx-f1 amounts (*Figure 3A*). Thus, inhibition was low in the presence of 0.625 µM Trx-f1, while a strong decrease was detected in 5 µM Trx-f1, revealing saturation with increasing Trx-f1 concentration.

2-CysPrx employs two Cys-residues, the peroxidatic Cys54 and the resolving Cys176, in the detoxification reaction of peroxides such as $H_2O_2$, alkylhydroperoxides and peroxinitrite. The dependency of the Trx oxidase activity of the 2-CysPrx on the presence of the catalytic thiols was tested by using site-directed mutated variants of 2-CysPrx (*König et al., 2013*) (*Figure 3B*). The variants C54S, C176S and also the hyperoxidation mimicking C54D were unable to inactivate FBPase, thus both thiols are needed. In a converse manner, the variant F84R which is compromised in its decamerization ability (*König et al., 2013*) but still acts as thiol peroxidase efficiently inactivated the FBPase in the complete inactivation assay.

The *in vitro* FBPase activity was simulated by use of a kinetic mathematical model (*Figure 3E*;*Figure 3—figure supplement 1*; Appendix 1) and the results compared with an experiment where oxidized 2-CysPrx was added to the enzyme assay at different concentrations (*Figure 3C*). The addition of 2.5, 5 or 10 µM 2-CysPrx$_{ox}$ progressively inhibited the turnover of FBP to Fru-6-P as indicated by absorption change. The simulation (*Figure 3D*) started with fully activated FBPase in the presence of 5 µM Trx-f1. Addition of 2-CysPrx$_{ox}$ at 2.5, 5, 10 or 20 µM led to rapid partial and concentrations-dependent inhibition of FBPase with high fit to the empirical data.

The NADPH-dependent malate dehydrogenase (MDH) is another established target of redox regulation in the chloroplast. MDH is activated by Trx-m if the reduction potential of the stroma turns highly negative for example in excess light. MDH activity was measured *in vitro* under similar conditions as the FBPase described above. Reductive pre-activation of MDH enabled the conversion of oxaloacetic acid to malate with concomitant oxidation of NADPH (*Figure 4A*). Addition of 2-CysPrx$_{ox}$ had no influence on MDH activity, showing full activation by DTT. The supplementation of the assay with various Trxs during pre-incubation enabled the inactivation of MDH after addition of 2-CysPrx$_{ox}$. Thus, Trx-m1 mediated complete MDH inhibition by 2-CysPrx$_{ox}$. The chloroplast drought-specific protein CDSP32 ranked second among the tested Trxs (*Figure 4B*). Thus, the overall order of inhibition efficiency was Trx-m1>CDSP32>Trx-f1>Trx-x>Trx-m4.

Inactivation of reductively activated proteins by oxidation is considered to be essential in photosynthesis upon lowering the photon flux density and darkening. The inhibition of CBC enzymes, ATP synthase and MDH prevents depletion of metabolites, suppresses futile cycles and long-lasting de-energization. MDH activity was measured in leaf extracts of high light-illuminated WT plants and the

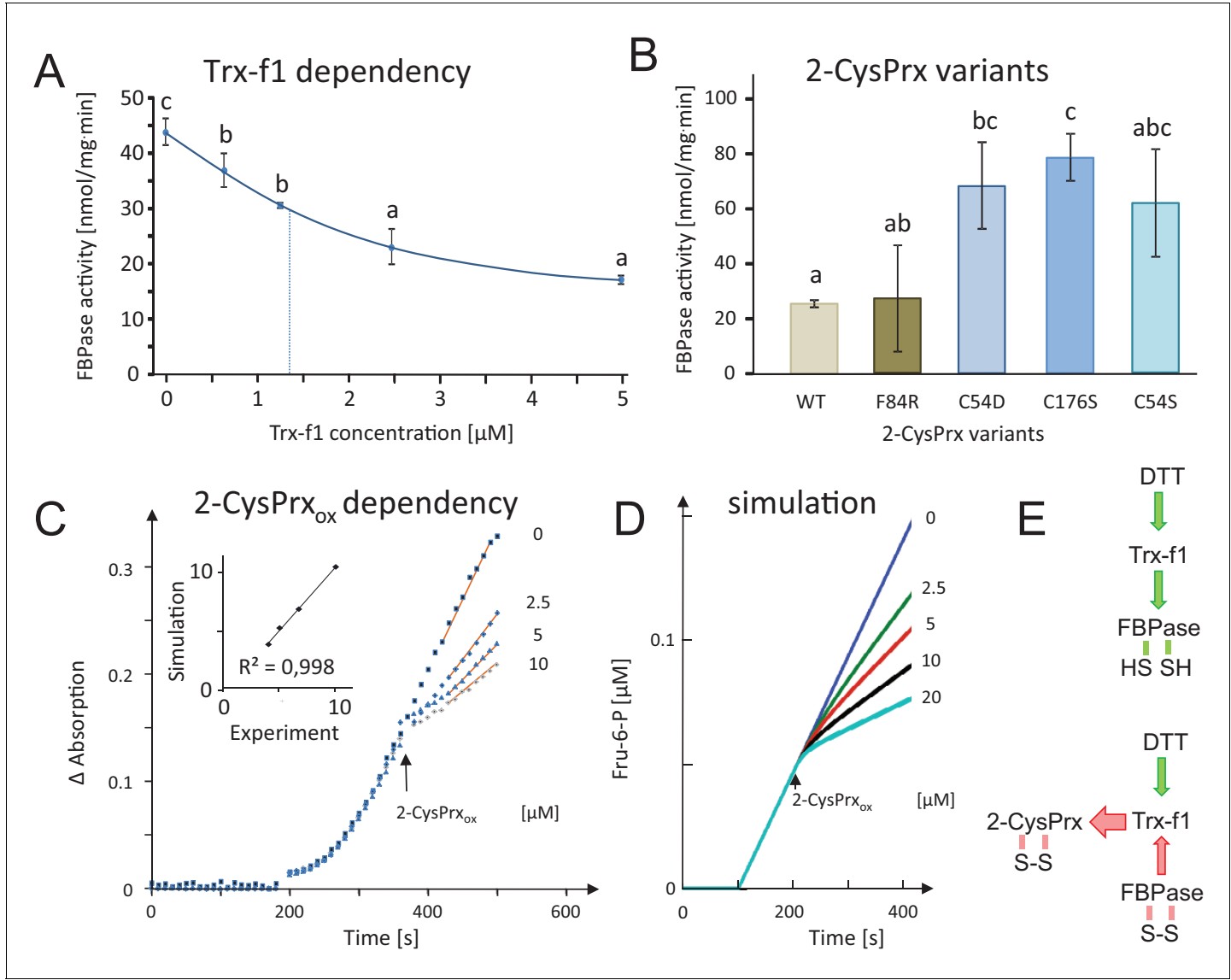

**Figure 3.** Dependency of FBPase inactivation on Trx-f1 concentration and functionality of the 2-CysPrxA, and mathematical simulation of the *in vitro* assay by kinetic modeling. (**A**) Dependency of FBPase activity on Trx-f1 concentration. At t = 0 min, Trx-f1 was added at concentration between 0 and 5 µM. The FBPase activity test was performed as in *Figure 2A*. Data are means ± SD of n = 3. Statistics were calculated using one-sided ANOVA followed by Tukey's HSD. Different letters mark significantly different groups. (**B**) Requirement of the functional thiol peroxidase for FBPase inhibition. The test contained 5 µM Trx-f1, final concentration of DTT was 500 µM and the 2-CysPrx$_{ox}$ variants were added at 5 µM concentration. The WT form of 2-CysPrx$_{ox}$ inhibited the FBPase by 75%, as did the F84R variant which is compromised in oligomerization. The variants devoid of the peroxidatic Cys in C54S and C54D, as well as the variant lacking the resolving Cys in C176S were ineffective in inhibiting the reductively activated FBPase. (**C**) Effect of adding different final 2-CysPrx$_{ox}$ concentrations to the ongoing FBPase activity test. Conditions were as described in *Figure 2E*. The traces are means of n = 3 determinations. (**D**) Mathematical simulation of the concentration-dependent effect of 2-CysPrx$_{ox}$ on FBPase activity. Addition of 0, 2.5, 5 and 10 µM, and also 20 µM 2-CysPrx$_{ox}$ was simulated (the effect of 20 µM 2-CysPrx$_{ox}$ could not be tested experimentally). The relative slopes after addition of 2-CysPrx$_{ox}$ in the experimental and theoretical analyses were plotted and gave a linear dependency with a regression coefficient of 0.998 (inset in *Figure 2C*). (**E**) Schematics of the measured and simulated pathways before (upper) and after addition of 2-CysPrx$_{ox}$ (lower scheme). Green arrows represent reductive activation, red arrows oxidative inactivation.

DOI: https://doi.org/10.7554/eLife.38194.007

The following source data and figure supplement are available for figure 3:

**Source data 1.** FBPase activity in the presence of different Trx-f1 concentrations.

DOI: https://doi.org/10.7554/eLife.38194.009

**Figure supplement 1.** Mathematical modeling and simulation of FBPase inactivation by 2-CysPrx$_{ox}$ in the enzyme assay.

DOI: https://doi.org/10.7554/eLife.38194.008

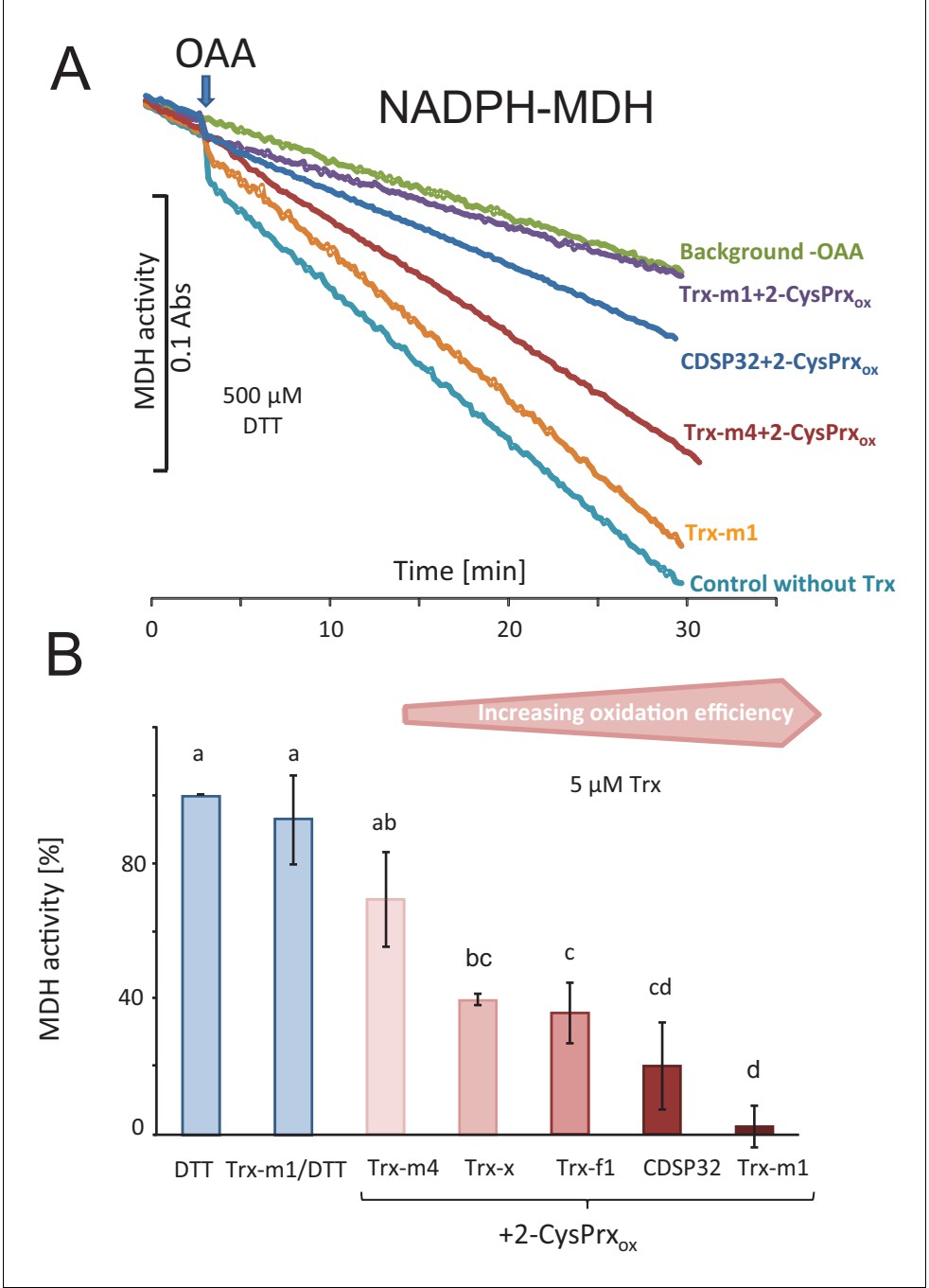

**Figure 4.** Trx-dependent inactivation of MDH by oxidized 2-CysPrx. (**A**) Spectrophotometric recording of MDH activity after addition of 2 mM oxaloacetic acid (OAA). The background was determined without addition of OAA. Addition of 10 µM Trx-m1 did not alter the turnover rate. Supplementation with combinations of different Trxs and 2-CysPrx$_{ox}$ inhibited the MDH to a varying degree. (**B**) Average inhibition of MDH activity in experiments as shown in (**A**). Data are means ± SD of n = 3–4. Statistics were calculated using one-sided ANOVA followed by Tukey's HSD. Different letters mark significantly different groups.

DOI: https://doi.org/10.7554/eLife.38194.010

The following source data is available for figure 4:

**Source data 1.** Values of the MDH activity test shown in *Figure 4B*.
DOI: https://doi.org/10.7554/eLife.38194.011

*2cysprxAB* insertion mutant in a time course following darkening (*Figure 5A*). MDH was rapidly inactivated in WT, while inhibition was delayed in *2cysprxAB*. The inactivation state was significantly different at 10 s after darkening of WT, C1, and C2 (*Figure 5A* and *Figure 5A—figure supplement 1*) compared to *2cysprxAB*. Residual activity of MDH after 10 s of darkness was 55.8% of the activity at 0 s in WT, 90.9% in *2cysprxAB* and 61.0% and 73.2% in C1 and C2, respectively. C1 and C2 represent plants expressing 2-CysPrxA under control of its endogenous promoter in the genetic background of *2cysprxAB* (*Figure 5—figure supplement 2*)

Ribulose-5-phosphate kinase (PRK) was chosen as another reductively activated CBC enzyme which catalyzes the committed step of ribulose-1,5-bisphosphate generation for RuBP carboxylation/oxygenation. The time course of PRK activity was followed during a light-dark transition and revealed deactivation in WT with progression of darkness while the activity remained unchanged in *2cysprxAB* within the time period of 300 s (*Figure 5B*). Both complemented lines C1 and C2 showed a degree of inhibition after 300 s like WT (*Figure 5—figure supplement 3*)

To address the more global effect of lacking 2-CysPrx, insoluble acid hydrolysable carbohydrates were measured during a day time course (*Figure 5C*). The amounts of insoluble carbohydrates increased 2.6-fold during the light phase and declined during the night in WT. Carbohydrate amount was 40% higher in *2cysprxAB* at the end of the dark phase and reached a similar amount at the end of the light phase as in WT. Mobilization in the dark phase started with a delay.

Ferredoxin is the central electron distributor transferring electrons from the photosynthetic electron transport chain (PET) to consuming metabolic reactions such as Fd-dependent NADPH reductase, Fd-nitrite reductase, Fd-sulfite reductase, Fd-Trx-reductase and others. The re-oxidation rate of Fd was determined in leaves from WT (*Figure 6A*), *2cysprxAB* (*Figure 6B*) and the complemented lines C1 and C2 (*Figure 6—figure supplement 1*) using the near infrared kinetic LED spectrometer (NIR-KLAS-100). Dark adapted leaves were illuminated with a short 1.5 s light pulse of 162 µmol quanta $m^{-2}$ $s^{-1}$ and then darkened again. During this short light period, the Fd pool was photoreduced.

The reoxidation of the Fd pool was followed as difference between the NIR absorption of the leaf at 785 and 840 nm (*Klughammer and Schreiber, 2016*). In WT plants, reoxidation occurred with a half-life time of $416 \pm 81$ ms, while Fd reoxidized with a half-life time $t_{50}$ of $120 \pm 35$ ms in *2cysprxAB* (n = 5, m $\pm$ SD) resulting in a very stable increase in oxidation rate by a factor of 3.5 if 2-CysPrx is missing. This decrease in $t_{50}$ indicates that the oxidative inactivation of electron-consuming reactions was effective in WT but markedly delayed in plants lacking 2-CysPrxAB. The fast decay rate in *2cysprxAB* was fully reversed to WT levels in the complemented lines C1 and C2 (*Figure 6—figure supplement 2*). The values were $408 \pm 56$ ms for C1 and $386 \pm 47$ ms for C2.

The redox state of 2-CysPrx was assessed during the light-dark-transition. In non-reducing gel separations of N-ethylmaleimide-blocked samples, the 2-CysPrx monomer at about 22 kDa corresponds to the reduced fraction, the dimer at 44 kDa to the fully or partially oxidized form. In the light, only about 40% of total 2-CysPrx was reduced. This fraction increased upon darkening and then decreased during extended darkness (*Figure 6C*). A considerable portion of the 2-CysPrx was oxidized both in the light and in the dark with transient variation during light transitions which might indicate the involvement in oxidation of Trx and Trx-dependent targets.

The previous data are in line with less efficient inactivation of metabolism in *2cysprxAB* compared to WT. This hypothesis was further scrutinized by a growth experiment with WT, *2cysprxAB*, and complemented lines C1 and C2 assuming that the slow inactivation of metabolism in *2cysprxAB* might be advantageous under certain fluctuating light conditions with short light pulses. WT is known to outperform *2cysprxAB* (*Figure 6D*) under normal growth conditions with continuous light during the day. This advantage was challenged under fluctuating light. While WT grew better in continuous short day illumination, this changed when WT and *2cysprxAB* grew in 80 s L/10 s H. Here, growth of *2cysprxAB* slightly surpassed growth of WT (*Figure 6D*).

The difference was less pronounced in 40 s L, 5 s H-cycles and in 40 s darkness, 5 s H, where the WT showed severe damage after 3 weeks, while the damage was less in *2cysprxAB* (*Figure 7*). Another difference was seen in continuous light without dark period. Rosette growth was rather similar. However, WT plants accumulated anthocyanins in older leaves while *2cysprxAB* remained green in all leaves of the rosette, indicating additional effects of the lack of 2-CysPrx on plant development and metabolism apart from being involved in the adjustment of metabolism during light shifts and light/dark transitions. *Table 1* depicts the growth data and the ratios of WT/*2cysprxAB* when grown

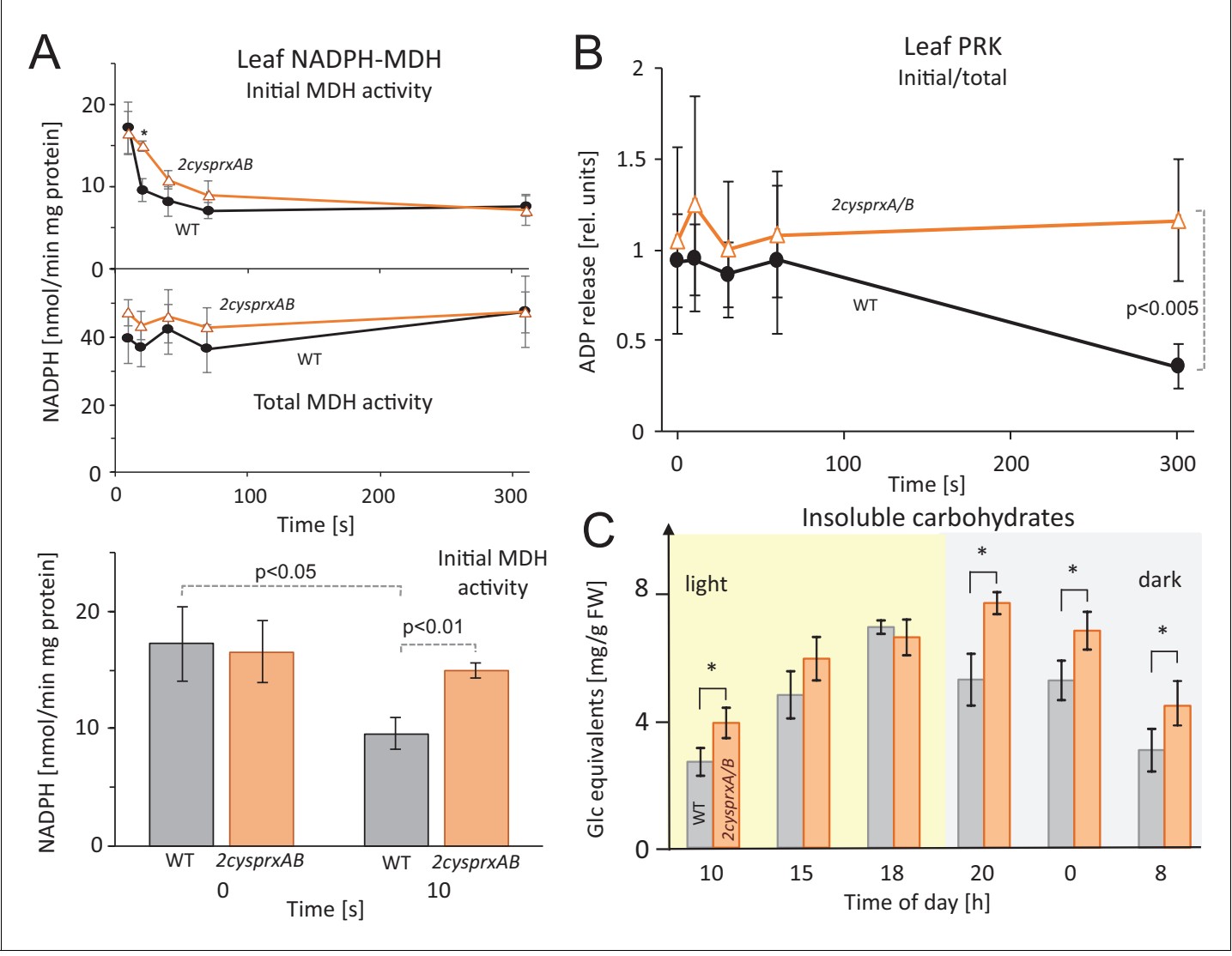

**Figure 5.** Inhibition of NADPH-MDH and ribulose-5-phosphate kinase in leaves upon light-dark transitions, and non-soluble sugar contents in WT and *2cysprxAB* during a 24 hr day-night cycle. (**A**) MDH activity during a light-dark transition: WT and *2cysprxAB* plants were exposed to 650 μmol quanta m$^{-2}$ s$^{-1}$ for 30 min and then darkened at t = 0 s. Proteins were rapidly extracted prior to darkening and after different time points in the dark as indicated. Initial (upper figure) and total MDH activity (lower figure) after full activation with 20 mM DTT were determined. Inactivation of MDH was slightly delayed in *2cysprxAB* than in WT. This is shown for the time point 10 s. Data are means ± SD from n ≥ 5 determinations. (**B**) Ribulose-5-phosphate kinase (PRK) activity during a light-dark transition plotted as ratio of initial to total activity. WT and *2cysprxAB* plants were illuminated with 650 μmol quanta m$^{-2}$s$^{-1}$ for 30 min and darkened as above. Initial and total PRK activities were determined and are represented as ratio. Initial inactivation was insignificantly delayed in *2cysprxAB*, but while PRK in WT continued to be inactivated until 5 min, the PRK activity essentially remained unaltered and became significantly different at t = 300 s (n = 6). Complemented lines behaved like WT (supplements). (**C**) Changes in insoluble carbohydrates during a 24 hr day-night cycle. Leaves of WT and *2cysprxAB* plants were harvested at 10 o'clock (1 hr after start of light phase), 15, 18, 20 (1 hr after end of light phase), 0 and 8 (1 hr before end of dark phase). Insoluble acid hydrolysable carbohydrates were quantified in the washed sediment of leaf homogenates using the Anthrone reagent following 1 hr boiling in sulfuric acid. Data are means ± SD of n = 3 extracts of different plants with three technical replicates each in Anthrone quantification. Statistics was calculated using one-sided ANOVA followed by Tukey's HSD, while * = p ≤ 0.05.

DOI: https://doi.org/10.7554/eLife.38194.012

The following source data and figure supplements are available for figure 5:

**Source data 1.** Altered MDH and PRK activities in *2cysprxAB* mutants after light-dark transitions.
DOI: https://doi.org/10.7554/eLife.38194.016
**Figure supplement 1.** Genotyping and protein detection in *2cysprxAB* and two independent 2-CysPrxA-complemented lines (C1 and C2).
DOI: https://doi.org/10.7554/eLife.38194.013
*Figure 5 continued on next page*

*Figure 5 continued*

**Figure supplement 2.** PRK activity in the complemented lines C1 and C2 in comparison to WT and *2cysprxAB*.
DOI: https://doi.org/10.7554/eLife.38194.014
**Figure supplement 3.** Simulation of redox change of FBPase upon darkening.
DOI: https://doi.org/10.7554/eLife.38194.015

in continuous and fluctuating light. *2cysprxAB* performed least in continuous light and best in 80 s low light and 10 s high light. The growth advantage of *2cysprxAB* in fluctuating light of 80 s low light (L) and 10 s high light (H) was lost after complementation.

## Discussion

The Trx-dependent reductive pathway of redox regulation participates in the control of most cellular processes (*Buchanan, 2016*). However, an open question concerns the mechanism of Trx re-oxidation which is employed to inactivate once reduced target proteins. Only the controlled interplay of reduction and oxidation enables efficient and fine-tuned regulation. The central role of $H_2O_2$ in tuning development and acclimation is generally accepted, but by which mechanism comes $H_2O_2$ into play? The properties of the oxidant(s) must meet several criteria in particular concerning the specificity, the thermodynamics including midpoint redox potential and sufficient amounts of oxidant buffer. The results from this study show that the chloroplast 2-CysPrx functions as peroxide-dependent thioredoxin oxidase. In the following, we discuss the three criteria and then the more general implications.

### Specificity of oxidation

The chloroplast thiol network is hierarchically structured. High priority for activation and likely also for inactivation has the γ-subunit of the F-ATP synthase (*Gütle et al., 2017*). Tight control is needed to initiate ATP synthesis as soon as the light-driven proton motive force (PMF) is generated, and also to inhibit the ATP synthase upon lowering or extinguishing the incident light in order to avoid ATP hydrolysis upon dissipation of the PMF (*Mills and Mitchell, 1982*). These authors proposed the existence of an oxidation system in the stroma that oxidizes the γ-subunit of the F-ATP synthase in the dark. Thiol-controlled activation of CBC enzymes such as FBPase, SBPase and PRK is of second priority in order to initiate carbon assimilation and high rate energy drainage. Thiol-dependent activation of NADPH-malate dehydrogenase comes third since the malate valve should only drain reductive power to the cytosol if the CBC fails to consume the available energy provided by the PET (*Backhausen et al., 1994*). It is accepted that this specificity of reductive activation is realized by the complex Trx system of the chloroplast (*Thormählen et al., 2017*).

In contrast to activation, the inactivation has scarcely been studied. Direct oxidation of the target proteins by $H_2O_2$ neither can provide specificity nor adequate prioritization. The Trxs that are instrumental in activating the targets, and in addition the Trx-like proteins such as ACHT1-4, provide a platform which could assist in mediating specificity and prioritization. This is seen in the results from the *in vitro* inactivation assays. Among the tested Trxs, FBPase was efficiently inactivated by Trx-f1 only. In case of MDH, Trx-m1 was most effective, but CDSP32 and Trx-f1 and Trx-m4 also inactivated MDH with lower efficiency. Apparently, the efficiency for inactivation of FBPase and MDH revealed the expected Trx specificity. Thus, complete switch off of MDH was achieved by the combination of Trx-m1 and 2-CysPrx$_{ox}$, while the FBPase was not entirely inhibited even in the presence of the preferred Trx-f1 under the chosen conditions. The simulation of the enzyme assay in a mathematical model showed that the 2-CysPrx-dependent inactivation of FBPase was in line with the reported kinetic constants and redox potentials.

Specificity in reoxidation could also arise from the five different thiol peroxidases targeted to the plastids, namely in addition to 2-CysPrxAB PrxIIE, PrxQ and glutathione peroxidase-like 1 and 7 (*Dietz, 2016*). The combinatorial network of about 20 plastidial Trxs and Trx-like proteins with five thiol peroxidases provide a framework for tuned activation and inactivation dependent on interaction ability, concentration and redox potentials.

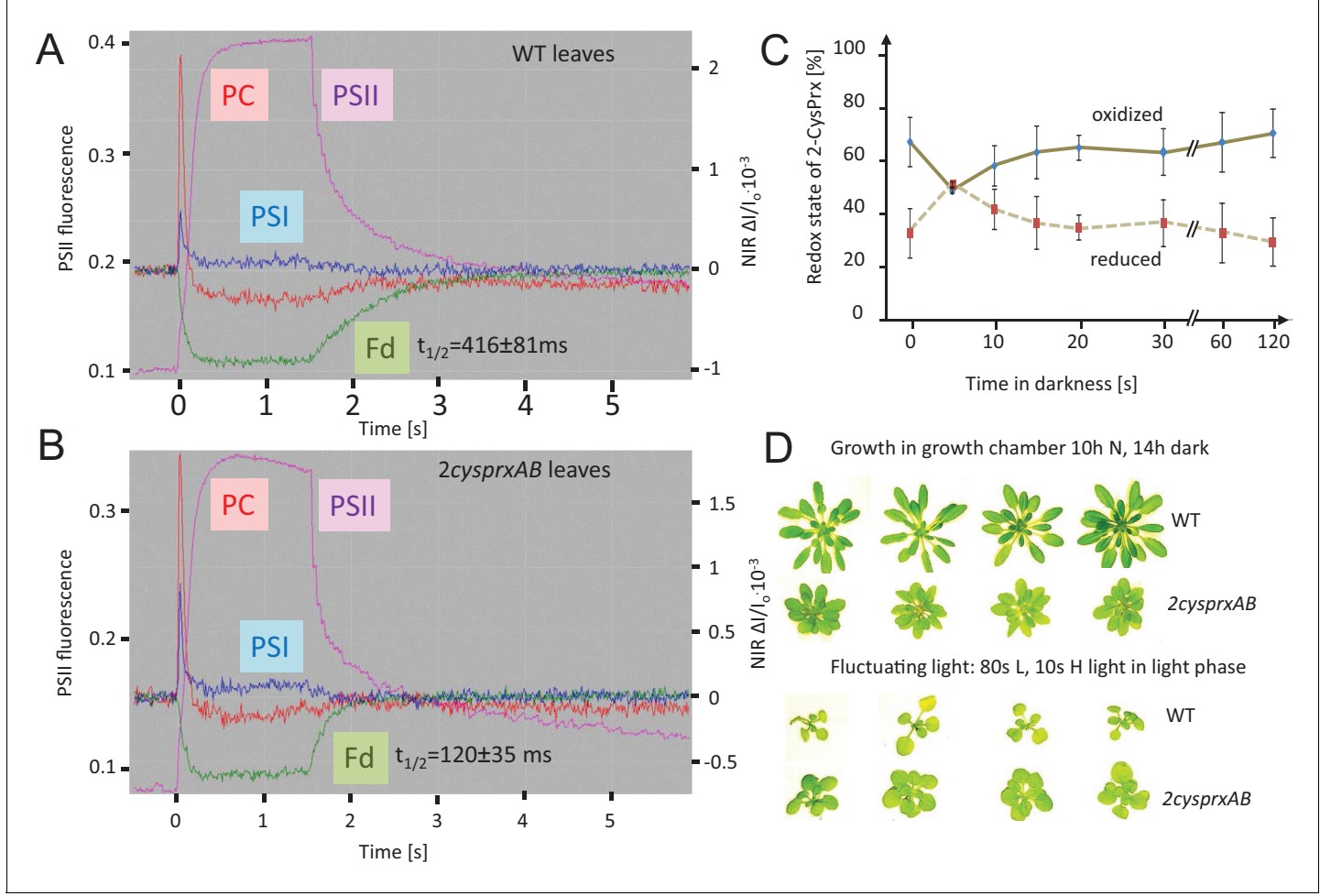

**Figure 6.** Reversal of light-induced changes of photosynthetic parameters. (**A**) WT and (**B**) *2cysprxAB* plants were acclimated to darkness. Fluorescence and NIR absorption changes were recorded with the NIR-KLAS-100. Chlorophyll-a fluorescence from photosystem II (PSII, violet trace) (left ordinate) and redox changes from photosystem I (PSI, blue), plastocyanin (PC, red) and ferredoxin (Fd, green) (right ordinate) were measured during a fast kinetics analysis consisting of 1.5 s illumination with 162 µmol quanta $m^{-2}$ $s^{-1}$ followed by darkening. The analysis of the fluorescence and absorption decay showed slower half-life times for WT than for *2cysprxAB* plants. The figure shows recordings typical for all analysis with n = 5 measurements. Complemented lines behaved like WT (supplement). (**C**) Redox state of the 2-CysPrx *ex vivo* during a light dark transition. WT plants were acclimated to 650 µmol quanta $m^{-2}$ $s^{-1}$ for 30 min (t = 0 s) and then darkened. Samples were harvested (t = 5 s-120 s) and blocked with 100 mM N-ethylmaleimide, separated by SDS-PAGE, immunodecorated with anti-2-CysPrx antibody and visualized by luminescence detection. The band intensities on shortly exposed films were quantified by ImageJ. Data are means ± SD of n = 4 experiments. (**D**) WT and *2cysprxAB* plants were grown in normal (N) light for 42 d (upper row) or for 14 d in growth light followed by 21 d in fluctuating light with 80 s L (20 µmol quanta $m^{-2}$ $s^{-1}$)/10 s H (800 µmol quanta $m^{-2}$ $s^{-1}$) during the 10 hr light phase (lower row). Growth of *2cysprxAB* was inhibited relative to WT in normal light, but the relative growth performance was reversed in fluctuating light. Four plants are shown for each growth condition and genotype.

DOI: https://doi.org/10.7554/eLife.38194.017

The following figure supplements are available for figure 6:

**Figure supplement 1.** Fluorescence and NIR absorption changes of the complemented lines C1 and C2 measured with the NIR-KLAS-100.

DOI: https://doi.org/10.7554/eLife.38194.018

**Figure supplement 2.** Half-life time of Fd reoxidation in WT and *2-cysprxAB* as well as *2cysprxAB* complemented with 2-CysPrxA (C1 and C2).

DOI: https://doi.org/10.7554/eLife.38194.019

**Figure supplement 3.** Exemplary immunoblot with leaf extracts using anti 2-CysPrx antiserum.

DOI: https://doi.org/10.7554/eLife.38194.020

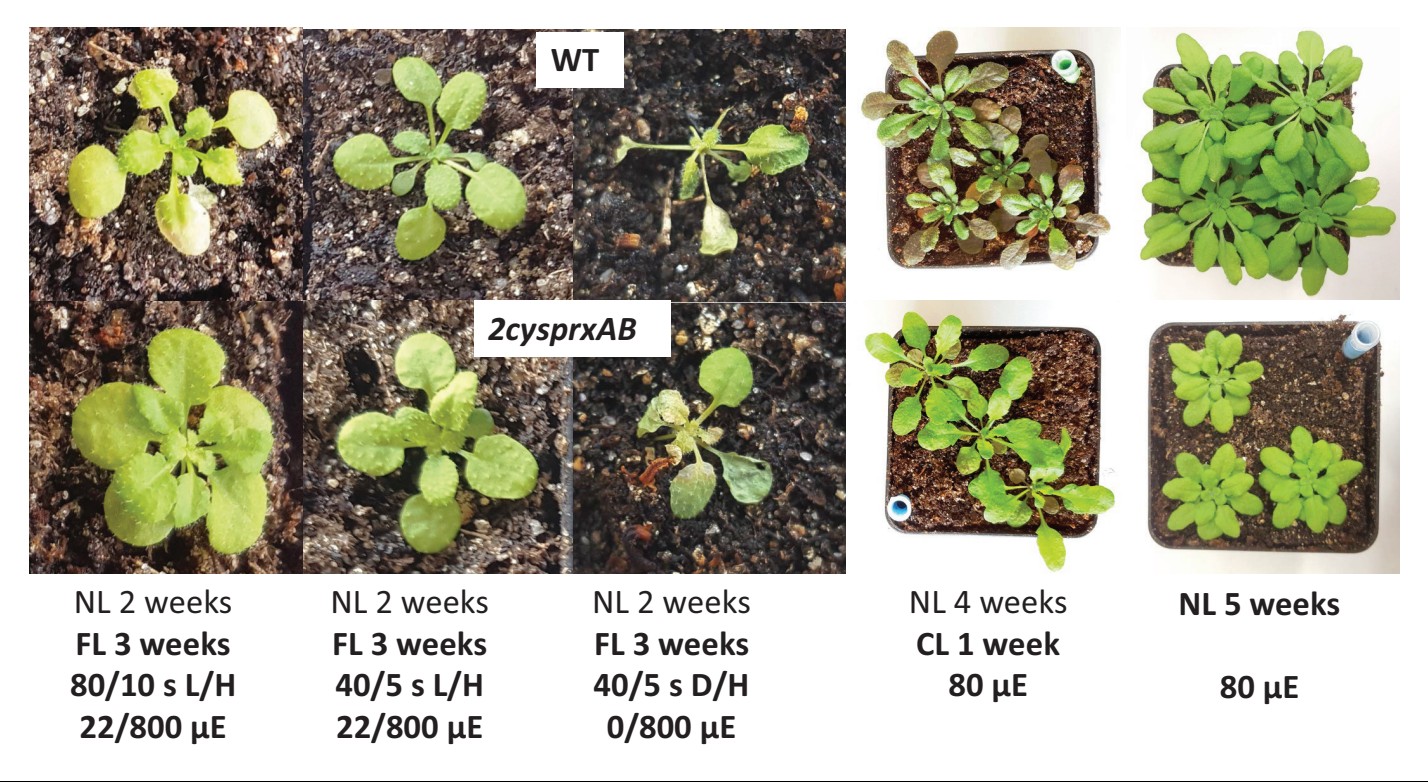

**Figure 7.** Growth phenotype of WT and *2cysprxAB* in different light regimes. WT (upper image) and *2cysprxAB* (lower image) were grown in five different light regimes as indicated, namely from left to right: (i) fluctuating light (FL) for 3 weeks consisting of 80 s/10 s L/H with 22/800 μmol quanta m$^{-2}$ s$^{-1}$ (as in *Figure 6D*). The daily light cycle was maintained at 14 hr dark phase/10 hr light phase. (ii) 3-week FL with 40 s/5 s L/H with 22/800 μmol quanta m$^{-2}$ s$^{-1}$. (iii) 3-week FL with 40 s/5 s darkness/H with 0/800 μmol quanta m$^{-2}$ s$^{-1}$. (iv) Growth in continuous light without dark phase for 1 week after 4 weeks of growth in the growth chamber. (v) 5-week-old plants grown in the growth chamber with 10 hr light/14 hr dark phase at 80 μmol quanta m$^{-2}$ s$^{-1}$. These growth experiments were independently performed three times with same result. Growth data are given in *Table 1* and *Figure 7— source data 1*.

DOI: https://doi.org/10.7554/eLife.38194.021

The following source data is available for figure 7:

**Source data 1.** Fresh weight for two fluctuating light schemes (40 s L/5 s H; 80 s L/10 s H) including complementation lines.

DOI: https://doi.org/10.7554/eLife.38194.022

## Thermodynamics of oxidation

The redox midpoint potential of 2-CysPrx was reported with −315 mV (*König et al., 2002*), while the E$^{'}_{M}$ of NADPH-MDH from sorghum and of spinach FBPase were determined with −330 mV, and

**Table 1.** Effect of fluctuating light on rosette fresh weight of *2cysprxAB* and WT.

Plants were grown in fluctuating light for 3 weeks. The program established the following cycles: 40 s L/5 s H, 80 s L/10 s H, 120 s L/15 s H; H corresponds to 800 μmol quanta m$^{-2}$ s$^{-1}$, L to 22 μmol quanta m$^{-2}$ s$^{-1}$ and normal growth light (80–100 μmol quanta m$^{-2}$ s$^{-1}$). Plants were harvested and analyzed for biomass accumulation depending on different light/dark schemes.

| Light cycle | WT [mg] | *2cysprxAB* [mg] | Ratio WT/*2cysprxAB* |
|---|---|---|---|
| constant | 372 ± 74 | 117 ± 20 | 3.19 |
| 40 s/5 s | 26 ± 5 | 22 ± 4 | 1.21 |
| 80 s/10 s | 21 ± 6 | 24 ± 5 | 0.87 |
| 120 s/15 s | 14 ± 3 | 11 ± 2 | 1.34 |

DOI: https://doi.org/10.7554/eLife.38194.023

−290 mV for spinach PRK (*Hirasawa et al., 1999*; *Hirasawa et al., 2000*). The $E'_M$ of spinach and pea Trx-f1 with −290 mV was less negative than the $E'_M$ of spinach Trx-m with −300 mV (*Hirasawa et al., 1999*). Thus, oxidized 2-CysPrx is thermodynamically able to withdraw electrons from MDH and FBPase through Trx. This electron drainage should be less efficient in the case of PRK because of its less negative $E'_M$. This was confirmed when estimating the initial vs. total PRK activity in leaves during a light-off experiment (*Figure 5*). PRK deactivated slowly in WT, while its activity remained unchanged in *2cysprxAB*. Thus, additional mechanisms must participate in the regulation of PRK. The inactivation was restored in the complemented lines C1 and C2 under control of the endogenous promoter proving the crucial role of 2-CysPrx in the deactivation process.

The rather negative redox midpoint potential of the regulatory thiols in NADPH-MDH may explain, why the difference between inactivation of MDH in WT and *2cysprxAB* was essentially quite small and only significant at 10 s after darkening. In addition, thiol oxidation is linked with metabolite control of the redox process and strongly affects activity of redox-regulated enzymes as recently reviewed by *Knuesting and Scheibe (2018)*.

The oxidized CP12 protein with disordered regions functions as chaperone to assemble an inactive supramolecular complex of glyceraldehyde-3-phosphate dehydrogenase and PRK (*Graciet et al., 2003*). In the regulatory scenario of the chloroplast, the photosynthetic light reaction feeds electrons via Fd into the Trx-system. The high free-energy change in electron transfer from Fd to Trx and target proteins efficiently drives their reductive activation. That is why in chloroplasts Trx reduction in the light is kinetically linked to Fd and not NADPH. Upon a drop in light intensity, the photosynthetic electron transfer to Fd ceases and Trxs are oxidized by drainage of electrons to 2-CysPrx. The transfer efficiency is not uniform, but shows significant specificity (i) through the Fd-dependent Trx reductase (*Yoshida and Hisabori, 2017*), (ii) between Trxs and targets and (iii) between Trxs and 2-CysPrx (this paper) (*Collin et al., 2003*). The efficiency of oxidation could be higher if the $E'_M$ of 2-CysPrx were higher (less negative). Since the oxidation reaction persists in the light and is counteracted by NADPH/NTRC (*Pérez-Ruiz et al., 2017*), reduction and oxidation of targets essentially comprises a futile cycle which should not consume too much energy. Thus, the evolutionary selection of a rather negative $E'_M$ for 2-CysPrx might be a characteristic which allows achieving this goal. But in addition, one should keep in mind that 2-CysPrx undergoes conformational changes and possibly other posttranslational modifications which might affect its $E'_M$ which should be explored in future work.

## The oxidant buffer pool size

The two highly identical isoforms of 2-CysPrxA and B represent very abundant proteins in the Arabidopsis stroma with a total concentration of more than 100 μM (*König et al., 2002*). A major function of 2-CysPrx is seen in its thiol peroxidase activity, participating in the ascorbate-independent water-water cycle and detoxification of Mehler reaction-derived $H_2O_2$ (*Dietz et al., 2006*; *Awad et al., 2015*) and, thereby, keeping the $H_2O_2$ concentration low. However, for most efficient thiol peroxidase function, the 2-CysPrx should ideally be fully or highly reduced. Surprisingly, the 2-CysPrx is highly oxidized under most conditions with less than 50% share of 2-CysPrx in the reduced form (*Pulido et al., 2010*) (*Figure 6C*). The rate of PET-dependent release of superoxide and subsequent $H_2O_2$ is considered to be low under growth conditions and may have regulatory function even under moderate stress (*Driever and Baker, 2011*). The oxidized fraction of $\geq$ 50% of the 2-CysPrx corresponds to a resting pool of 30–50 μM disulfide available for drainage of electrons from the Trx/Trx-like proteins and target pools. The 2-CysPrx redox state in the steady state underestimates the capacity for oxidation of target protein since the thiol-disulfide redox network is a dynamic flux system where $H_2O_2$ oxidizes and NTRC and Trx reduce the 2-CysPrx. In this context it is interesting that PrxIIE, the other soluble peroxiredoxin of the stroma has a midpoint redox potential of −288 mV (*Horling et al., 2003*), and thus could participate in oxidizing target proteins with less negative thiol redox potential.

## The regulatory context of metabolism

The effect of the 2-CysPrx$_{ox}$/Trx-system on certain CBC enzymes and the malate valve is evident from both our *in vitro* analyses and *in vivo* data including the reoxidation kinetics of Fd in leaves upon darkening. Interestingly, the regulatory impact of 2-CysPrx on the metabolic state of the

chloroplast goes far beyond carbon fixation and export of excess reducing power. The diurnal carbo-hydrate dynamics in *2cysprxAB* was disturbed. The carbohydrate amount in *2cysprxAB* reached its maximum in the early night phase and exceeded that of WT during the night and in the early day phase. Starch synthesis and degradation are subjected to redox regulation (*Santelia et al., 2015*). Trx-f1 contributes to activation of ADP-glucose pyrophosphorylase. Plants lacking Trx-f1 accumulate less starch and more soluble sugars in the light (*Thormählen et al., 2013*). The changes in diurnal carbohydrate pattern indicate that starch degradation is impaired in *2cysprxAB*. Enzymes involved in starch degradation also exhibit redox sensitive and regulatory thiols. However, degradation enzymes are reported to be more active in the reduced than oxidized state (*Santelia et al., 2015*), thus the exact role of 2-CysPrx in tuning the activity of starch turnover needs future analysis.

Other metabolic changes concern the accumulation of the aromatic amino acids phenylalanine and tryptophan which decreased in *2cysprxAB*, and the inhibited ability of *2cysprxAB* to synthesize anthocyanins in high light (*Awad et al., 2015*; *Müller et al., 2017*). Phenylalanine synthesis proceeds in the chloroplast via arogenate (*Jung et al., 1986*) and phenylalanine availability determines antho-cyanin synthesis (*Chen et al., 2016*). Redox-dependent regulation of the involved metabolic path-ways still needs to be explored. However, the *2cysprxAB* phenotype of compromised anthocyanin accumulation resembles the phenotype observed in ascorbate-deficient mutants which was linked to regulation of genes involved in anthocyanin synthesis (*Page et al., 2012*).

The contribution of 2-CysPrx to redox regulation appears particularly important in low light, in darkness and in fluctuating light. Thus, the phenotype of inhibited rosette growth with more round shaped leaves and short petioles (*Pulido et al., 2010*; *Awad et al., 2015*) was mostly reversed to WT phenotype in continuous light and high light (*Figures 6* and *7*). Fine-tuning of metabolic activi-ties by 2-CysPrx-dependent oxidation counteracts reductive activation and only the proper balance between reduction and oxidation allows for optimized acclimation to the prevailing environmental conditions. High rate photosynthesis in high light requires little oxidative drainage of regulatory elec-trons. At lower light intensities, the ratio of oxidation to reduction should increase for down-regula-tion of enzyme activities. If this oxidative counterbalance by 2-CysPrx is missing, enzymes maintain higher activity than needed, and this causes metabolic imbalances, sustains futile cycles and compro-mises growth performance of *2cysprxAB*.

The disadvantage of lacking 2-CysPrx converts to an advantage if the fluctuating light regime adopts a particular cycle frequency where deactivation by 2-CysPrx$_{ox}$ in WT proceeds too fast to exploit the short subsequent light phase (*Figures 6* and *7*). In a converse manner, the delayed regu-lation in *2cysprxAB* allows for exploiting the short high light phase for carbon assimilation. This sce-nario explains, why *2cysprxAB* plants outperform WT plants in the 5 and 10 s H-light cycle but turns again to a disadvantage with 15 s longer light phase.

The kinetic model of the *in vitro* enzyme test simulated the effect of oxidized 2-CysPrx on FBPase activity extremely well. Thus, the correlation between the experimental data and the simulated data after supplementation with 0, 2.5, 5 and 10 μM 2-CysPrx$_{ox}$ gave a linear dependency with a regres-sion coefficient of 0.998. In a converse manner, the simulation of the light-dark transition did not realize full inhibition of FBPase in the simulated night. In this context, it is important that other bio-chemical parameters such as FBP- and Fru-6-P-levels, Mg$^{2+}$, Ca$^{2+}$ and pH affect the redox sensitivity of FBPase (*Chardot and Meunier, 1991*). FBP is present at very low amounts in chloroplasts of dark-ened leaves (*Dietz and Heber, 1984*). The drop in FBP concentration is suggested to ease FBPase deactivation. Ca$^{2+}$ fluxes into the stroma upon darkening deactivate carbon assimilation (*Hochmal et al., 2015*). Stromal pH and Mg$^{2+}$ concentrations decrease upon darkening (*Ishijima et al., 2003*) and assist in enzyme regulation (*Minot et al., 1982*). These additional regulat-ing parameters likely must be considered to fully simulate the CBC enzyme deactivation *in vivo*.

The results from this study strongly support the conclusion that the 2-CysPrx functions as Trx-oxi-dase *in vivo* which mediates the inactivation of Trx-dependent target proteins. This mechanism is needed to effectively inactivate or downregulate the assimilation pathways linked to the photosyn-thetic electron transport chain in darkness and upon a drop in photosynthetic active radiation. Beyond controlling the CBC and the malate valve, other metabolic processes such as diurnal starch accumulation and non-photochemical quenching are also affected by this pathway, and thus 2-CysPrx appears to be a global player in tuning the redox regulatory network of the chloroplast.

## Materials and methods

### Growth of *Arabidopsis*

This work used *A. thaliana* Col-0 as wildtype control and T-DNA insertion lines in the Col-0 background. The *2cysprxAB* double insertion line was a gift of Prof. M. Müller (Würzburg University) and was published and characterized by *Awad et al. (2015)*. Both 2-cysprx genes are mutated. The parent lines were GK-295C05 (2-Cys PrxA, At3g11630) and SALK_017213C (2-Cys PRXB, At5g06290) which were crossed and selected for homozygocity. The *cyp20-3* insertion line (SALK_054125) was selected and characterized as described by *Müller et al. (2017)*. Plants were grown on soil in 10 hr light at 21°C and 14 hr darkness at 18°C with 50% relative humidity. After stratification for 2 d in the dark at 4°C plants were grown at 80 μmol quanta m$^{-2}$ s$^{-1}$ (normal light: N) for isolating chloroplasts after 56 d and also for enzyme assays and NIR KLAS100 analysis after 28 d. Growth studies utilized a fluctuating light regime to mimic sun and shade in a natural environment and to explore growth advantages of plants with missing 2-CysPrx. To this end both lines were grown in N for 14 d or as indicated and then transferred to a home-built programmable LED-based light chamber with fluctuating light consisting of low light (L)-phases with 22 μmol quanta m$^{-2}$ s$^{-1}$ and high light (H) phases with 800 μmol quanta m$^{-2}$ s$^{-1}$. The fluctuating light consisted of 80 s L/10 s H or 40 s L/5 s H for 28d or as indicated. Control plants were kept in N. Diurnal light-dark cycles were as described above.

### *Agrobacterium tumefaciens*-mediated transformation of plants

Generation of plasmids containing the endogenous promotor region and full length CDS of 2-CysPrx gene At3g11630 was done by conventional cloning and PCR methods. *A. thaliana* WT RNA was used for synthesis of cDNA as template for CDS amplification in PCR using primers At2CPCDS-NcoIrev and At2CPCDS-XhoIfor) (*Supplementary file 1*). The promotor region was amplified using At2CPprm-XhoIrev and At2CPprm-EcoRIfor on genomic DNA. Both fragments were consecutively cloned into pJAN backbone between left and right border regions using restriction enzymes and ligation (promotor: EcoRI, XhoI; CDS: XhoI, NcoI). The pJAN vector is based on pPAM (GenBank: AY027531) and used for T-DNA binary vector systems in GV3103 pMP90RK with selection markers Amp for bacteria and Kan for plants. Freeze/heat-induced transformation, selection and cultivation were conducted with low-salt LB-media (10 g tryptone, 5 g yeast extract, 5 g NaCl per liter). For transformation a floral dipping protocol according to *Zhang et al. (2006)* was slightly modified. Thus, 100 μM acetosyringone were added to the induction media for increased flowering and enhanced bacterial infection. The dipping was repeated after 2 d to increase transformation rate. Cultivation of sterilized seeds on ½-MS selection plates was done for two weeks followed by growth on soil for seed production. This was repeated until the T3 generation in which gene insertion and protein levels were examined (*Figure 5—figure supplement 5*). For genotypisation of WT, gene primers At2-CysPrxgenF + At2-CysPrxgenR were used and should result in a 1160 bp band in WT only. T-DNA insertion validation was done with primers At2-CysPrxgenR + GKRBfor for a 479 bp band only in mutant lines. A control for complementation was achieved by using primers At2CPCDS-NcoIrev + At2-CPCDS-XhoIfor for full CDS being slightly shorter than genomic DNA with 801 bp in the complementation lines C1 and C2.

### Chlorophyll-a fluorescence kinetics in seedlings

Wildtype, *2cysprxAB* and *cyclophilin 20–3* mutants lines were sterilized, placed on phytogel-solidified half strength Murashige-Skoog medium with or without 0.5% sucrose, stratified for 2 d and then grown in a growth chamber with 8 hr light, 16 hr dark at 80 μmol quanta m$^{-2}$ s$^{-1}$ and 23°C. Plants lacking cyclophilin 20–3 (At3g62030) are T-DNA insertion lines obtained from SALK institute (SALK_054125). FlourCam (Photon Systems Instruments, Czech Republic, Drasov) was used to analyze chlorophyll a-fluorescence in 4 d and 7 d old seedlings. Settings were applied as follows: 30 min dark acclimation, measuring light on after 5 s, actinic light with 65 μmol quanta m$^{-2}$ s$^{-1}$ after 16.5 s. Actinic light was switched off after 76.5 s and the recording terminated at 94.5 s. Saturating light pulses of 900 μmol quanta m$^{-2}$ s$^{-1}$ were applied after 5, 25, 75 and 85 s.

## Fluorescence of PSII, NIR-analysis of P700, PC and Fd absorption changes

Photosynthetic parameters were analyzed with a NIR-KLAS-100 spectrophotometer (Waltz, Germany, Effeltrich) using near infrared absorption spectroscopy (NIR) to visualize redox states of ferredoxin (Fd), plastocyanin (PC), photosystem I (PSI), and fluorescence from photosystem II (PSII) in leaves. Dark adapted plants were taken at the end of their normal dark phase and maintained in darkness. Two leaves were excised and sandwiched abaxially to fit into the $1 \times 1$ cm detection window for double-sided exposure. Fast kinetics was recorded in a 6 s time window with 1.5 s actinic light pulse of 162 µmol quanta $m^{-2}$ $s^{-1}$ to activate photosynthesis. Redox kinetics of the parameters was recorded for 4.5 s. Settings and device output were adjusted according to *Klughammer and Schreiber (2016)* and *Schreiber and Klughammer, 2016*.

## Gene cloning

The sequences encoding the mature proteins without targeting sequence of the Trx-f1, -m1, -m4, -x, CDSP32, 2-CysPrxA and FBPase from *A. thaliana* were amplified from leaf cDNA. Trx-f1, -m1, -m4, -x, CDSP32 and FBPase were cloned into the *NdeI* and *BamHI* restriction sites of pET15b (Novagen). 2-CysPrxA was cloned into the *NdeI* and *EcoRI* restriction sites (underlined in the primers) of the pET28a (Novagen) using the primers described in *Supplementary file 1*. Sequences were verified by DNA sequencing.

## Production and purification of recombinant proteins

The recombinant plasmids were introduced into the *E. coli* NiCo21 (DE3) strain (NEB). The bacteria were grown at 37°C, and protein production was induced by adding 100 µM isopropyl-β-D-thiogalactoside in the exponential phase. The bacteria were harvested by centrifugation at 5000 rpm for 20 min and then resuspended in buffer A (50 mM phosphate buffer, pH 8.0, 10 mM imidazole, 250 mM NaCl) for Trx-f1, -m1, -m4, -x, CDSP32 and FBPase, and buffer B (50 mM phosphate buffer, pH 8.0, 10 mM imidazole, 250 mM NaCl, 40 µM of β-mercaptoethanol) for 2-CysPrxA. Cells were disrupted by sonication, and debris was sedimented at 13,000 rpm for 30 min. The soluble fraction was loaded onto His-Select nickel affinity column (His-Select HF Nickel Affinity, ROTH) equilibrated with buffer A. Proteins were eluted with 50 mM phosphate buffer, pH 8.0, 250 mM imidazole, and 250 mM NaCl. Purified proteins were concentrated and dialyzed against 50 mM phosphate buffer, pH 8.0. Final purity was checked by 15% SDS-PAGE. Protein concentrations were determined spectrophotometrically using molar extinction coefficients at 280 nm of 17085 $M^{-1}$ $cm^{-1}$ for Trx-f1, 21095 $M^{-1}$ $cm^{-1}$ for Trx-m1, 19650 $M^{-1}$ $cm^{-1}$ for Trx-m4, 11585 $M^{-1}$ $cm^{-1}$ for Trx-x, 12170 $M^{-1}$ $cm^{-1}$ for CDSP32, 20065 $M^{-1}$ $cm^{-1}$ for 2-CysPrxA and 38195 $M^{-1}$ $cm^{-1}$ for FBPase. For oxidation, the recombinant 2-CysPrx was incubated in the presence of 1 mM $H_2O_2$ for 45 min and dialysed over night at 4°C with several changes of dialysis buffer (30 mM Tris-HCl, pH 8).

## Stroma isolation

Leaves were harvested from more than 20 plants previously darkened for starch degradation and blended in homogenization buffer (300 mM sorbitol, 20 mM Tricine, 5 mM EGTA, 5 mM EDTA, 10 mM $NaHCO_3$, 2 mM ascorbate and 1% bovine serum albumin; pH 8.4 with NaOH)(*Ströher and Dietz, 2008*). After filtering through 4 layers of gauze and nylon mesh the suspension was centrifuged, and the pellet re-suspended and spun again. Finally, chloroplasts were re-suspended in buffer containing 300 mM sorbitol, 30 mM KCl, 1 mM $MgCl_2$, 0.2 mM $KH_2PO_4$, 2 mM ascorbate and 50 mM Hepes-NaOH pH 7.6. The suspension was stored at −80°C, thawed and centrifuged for 2 hr to get stromal protein as supernatant.

## *In vitro* FBPase and MDH activities

The fructose 1,6-bisphosphatase (FBPase) activity was measured spectrophotometrically with the Shimadzu (UV-2401PC) at 340 nm (25°C). Stromal protein (100 µg) was preincubated in 30 mM Tris-HCl, pH 8.0, and 5 mM $MgSO_4$ with DTT, with or without thioredoxin (Trx, 5 µM) as indicated. The preincubation solution (500 µl) was added to 500 µl reaction mix (0.1 mM $NADP^+$, 30 mM Tris-HCl pH 8.0, 5 mM $MgSO_4$, glucose-6-phosphate dehydrogenase and phosphoglucoisomerase (0.5 U each)) and oxidized 2-CysPrxA (final concentration: 5 µM 2-CysPrx$_{ox}$). After recording the baseline

for 3 min the assay was initiated by adding fructose 1,6-bisphosphate (0.6 mM FBP). In a second test with exclusively recombinant proteins, purified FBPase protein was prepared and tested in the same assay. To this end purified FBPase (final concentration: 2 μM) was activated with 1 mM DTT in 500 μl preincubation mix also containing Trx-f1 (final concentration 5 μM). The preincubation mix was added to 500 μl reaction mix containing all components as above. At the time point as indicated, 5 μM of oxidized and dialyzed recombinant 2-CysPrxA was added to the reaction medium.

The malate dehydrogenase (MDH) test was conducted spectrophotometrically using the Cary 300 Bio UV-Visible (Varian, Middelburg, The Netherlands). The stromal protein (150 μg) was preincubated in buffer A (30 mM Tris-HCl, pH 8.0, 0.5 mM DTT, 0 or 10 μM Trx). Preincubation solution (500 μl) was mixed with 500 μl buffer B (30 mM Tris-HCl, pH 8.0, 0.4 mM NADPH and with or without 5 μM $2CysPrxA_{ox}$). After recording the baseline at 340 nm, the assay was started by adding 2 mM oxaloacetic acid (OAA) as substrate.

## Enzyme activities in leaf extracts

For the MDH activity assay intact plants were exposed to high light of 650 μmol quanta $m^{-2}$ $s^{-1}$ for 30 min. After 0, 10, 30, 60 and 300 s in darkness, leaves were immediately frozen in liquid nitrogen. After addition of 450 μl 30 mM Tris-HCl, pH 8.0, to 100 mg pulverized leaf material, samples were vortexed for 20 s and centrifuged at 16,000 x g for 90 s. In order to measure the initial MDH-activity 200 μl of supernatant was added to a quartz cuvette containing 30 mM Tris-HCl, pH 8.0, 0.1 mM NADPH and 2 mM OAA. NADPH oxidation was measured at 340 nm (Cary 300) at 25°C for 4 min. Total MDH-activity was determined after preincubation with 20 mM DTT for 30 min at RT. Background reactions were subtracted using an assay without OAA.

For PRK activity, pulverized leaf material (25 mg) from light-dark transitions was added to 450 μl PRK assay buffer (30 mM Tris-HCl pH 8.0, 1 mM EDTA, 40 mM KCl, 10 mM $MgCl_2$) and processed as above. To measure the initial PRK activity 50 μl of supernatant was added to a quartz cuvette containing 2 mM ATP, 2.5 mM phosphoenolpyruvate, 5 U/ml pyruvate kinase, 6 U/ml lactate dehydrogenase and 0.2 mM NADH. The reaction was started with 0.5 mM ribulose-5-phosphate and the activity was measured as a decrease in absorbance at 340 nm. The total PRK activity was quantified after preincubation with 20 mM DTT as above.

## FBPase redox state determination

100 μg of stroma was incubated under the conditions used for the FBPase activity assay. 500 μl of the reaction mixtures were stopped with one volume of 25% (w/v) trichloroacetic acid (TCA), and stored on ice for 1 hr. The mixtures were centrifuged at 13,000 rpm for 10 min and washed once with cold acetone. The pellet was resuspended in alkylation buffer 30 mM Tris-HCl pH 8.0 with 15 mM of $mPEG_{24}$-mal (methoxyl-PEG maleimide) (Thermo Fischer Scientific) and incubated in the dark at room temperature for 1 hr. The alkylated mixtures were precipitated with 25% TCA, washed with cold acetone and resuspended in a loading buffer containing 30 mM Tris-HCl pH 8.0, 2.3% SDS. The samples were separated in non-reducing SDS-PAGE (10%), transferred onto nitrocellulose membrane and probed with an anti-FBPase antibody which was kindly provided by Prof. Jean-Pierre Jacquot (University of Lorraine, Nancy, France).

## Analysis of soluble and insoluble carbohydrates

Plants for carbohydrate analysis were harvested at day times as indicated (one hour before and after the light phase and in the middle of the day or night). Plants grew in short day with 10 hr light with 80 μmol quanta $m^{-2}$ $s^{-1}$ at 21°C and 14 hr darkness at 18°C with 50% relative humidity. Carbohydrates were analyzed in 96-well plates according to *Leyva et al. (2008)* with minor modifications. Harvested WT and *2cysprxAB* leaves were grinded in liquid nitrogen and adjusted to 20–30 mg for optimal readings according to the linear calibration curves. After sedimentation at 13,000 rpm for 10 min the sediment was washed with phosphate buffer and boiled at 100°C for 1 hr in 3% hydrochloric acid to hydrolyze insoluble carbohydrates. After sedimentation at 13,000 rpm for 10 min, the supernatants were used as samples for insoluble carbohydrate determination. To 150 μl Anthron reagent (2 g $l^{-1}$ in 75% (v/v) sulfuric acid) 50 μl of sample or standard was added with gentle mixing prior to heating for 20 min at 100°C. Insoluble acid-hydrolysable sugar was determined at 630 nm in three independent replicates using a glucose standard curve.

## Redox state of 2-CysPrx in high light to dark shifts

Plants were grown in growth light for 28 d as described before. WT was acclimated to 650 µmol quanta m$^{-2}$ s$^{-1}$ for 30 min (t = 0 s) and transferred to darkness (t = 5 to 120 s). Plant material was grinded and extracted with 50 mM Tris-HCl, pH 8.0 containing 100 mM N-ethylmaleimide (NEM) to alkylate free thiols while existing disulfides are kept oxidized. Separation on non-reducing 1D-SDS-PAGE allowed discrimination of blocked monomers (22 kDa) and oxidized intermolecular homo-dimers (44 kDa). The following Western Blot with luminescence output was used in combination with ImageJ software to quantify protein band intensities using 1 or 2 µg of protein extract for optimal exposure. Values given are percentage shares of reduced (NEM-labeled) and oxidized (oxidized SDS-PAGE) 2-CysPrx at each time point with n = 4 replicates and ± SD.

## Kinetic modeling of the redox networks

Two models were implemented in Matlab. The first model was built to simulate the *in vitro* assay with pre-activation, Trx and FBPase and addition of oxidized 2-CysPrx at different concentrations. The parameters (concentrations, reaction constants) underlying the equations, the implementation of the redox potentials and the equations are given in Appendix 1 and 2. The second model was constructed to simulate the light-dark transition. In the presumed light phase, the fraction of reduced Fd was set to 50%. To mimic the light-to-dark transition after 200 s, the Fd and FTR pool were turned fully oxidized, assuming the efficient drainage of electrons into the various Fd-dependent pathways. The redox state of 2-CysPrx was clamped to 66% oxidized and 34% reduced form according to our own data (*Figure 6C*)

## Statistical analysis

Statistics was assessed using one-way analysis of variance (ANOVA) followed by post hoc analysis Tukey's HSD (Honest significance difference) test (p ≤ 0.05). Statistical results (significant difference) are indicated in the figures and represented by different letters. Statistical analyses were performed using IBM SPSS 20.0 software (IBM Corporation, Armonk, NY).

## Data availability

The authors declare that the data supporting the findings of this study are available within the article and the Supplementary Information files or are available upon reasonable request to the authors.

# Acknowledgements

The authors acknowledge support by the DFG (DI 346/14 and 17, SPP1710). We are grateful to Sergej Kakorin (Physical und Biophysical Chemistry, Bielefeld University) and Petra Lutter (Faculty of Biology, Bielefeld University) for help in kinetic modelling.

# Additional information

## Funding

| Funder | Grant reference number | Author |
| --- | --- | --- |
| Deutsche Forschungsgemeinschaft | Di346/14 | Karl-Josef Dietz |
| Deutsche Forschungsgemeinschaft | SPP1710 | Karl-Josef Dietz |

The funders had no role in study design, data collection and interpretation, or the decision to submit the work for publication.

## Author contributions

Mohamad-Javad Vaseghi, Investigation, Visualization, Performed the in vitro analyses of FBPase and MDH, the Trx dependencies, Determined the FBPase redox state; Kamel Chibani, Supervision, Investigation, Visualization, Writing—original draft, Writing—review and editing, Prepared recombinant

proteins, Assisted in performing the in vitro experiments and their interpretation; Wilena Telman, Investigation, Visualization, Writing—review and editing, Measured MDH and PRK activity ex vivo; Michael Florian Liebthal, Investigation, Visualization, Writing—original draft, Determined the 2-CysPrx redox state ex vivo, the carbohydrate levels, Prepared recombinant proteins, Performed A. tumefaciens-mediated transformation of plants as well as selection, cultivation and characterization, Performed the NIR KLAS100 and the fluctuating light experiments; Melanie Gerken, Formal analysis, Investigation, Visualization, Writing—original draft, Constructed the kinetic model for simulating the regulation kinetics; Helena Schnitzer, Sara Mareike Mueller, Investigation, Performed the chlorophyll a fluorescence analyses of plate grown plants, Cloned plasmids for A. tumefaciens-mediated transformation of plants; Karl-Josef Dietz, Conceptualization, Resources, Supervision, Funding acquisition, Writing—original draft, Project administration, Writing—review and editing

### Author ORCIDs
Mohamad-Javad Vaseghi (iD) http://orcid.org/0000-0002-4892-278X
Kamel Chibani (iD) http://orcid.org/0000-0003-3229-400X
Wilena Telman (iD) http://orcid.org/0000-0002-6336-486X
Michael Florian Liebthal (iD) http://orcid.org/0000-0003-3198-0358
Helena Schnitzer (iD) https://orcid.org/0000-0002-1027-1803
Karl-Josef Dietz (iD) https://orcid.org/0000-0003-0311-2182

### Decision letter and Author response
Decision letter https://doi.org/10.7554/eLife.38194.037
Author response https://doi.org/10.7554/eLife.38194.038

## Additional files

### Supplementary files
• Supplementary file 1. Primers used for cloning. for: forward primer, rev: reverse primer.
DOI: https://doi.org/10.7554/eLife.38194.024

• Transparent reporting form
DOI: https://doi.org/10.7554/eLife.38194.025

### Data availability
All data generated or analysed during this study are included in the manuscript and supporting files. Original and aggregated data are provided in the supplementary data file.

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

# Appendix 1

DOI: https://doi.org/10.7554/eLife.38194.026

## Modelling the inhibition of FBPase by oxidized 2-CysPrx in the presence of Trx-f1.

The model of the enzyme test consisted of Trx-f1, FBPase, 2-CysPrx and FBP as well as F6P. The variables of Trx-f1, FBPase and 2-CysPrx are represented in the oxidized and reduced form. *Appendix 1—table 1* gives the values for the starting condition and the simulated changes at t = 100 s, where addition of FBP as substrate was simulated and at t = 200 s where oxidized 2-CysPrx was added at varying concentrations.

**Appendix 1—table 1.** Values of variables used for modeling the enzyme assay.

| Component | Start value [$\mu M$] | Addition $_{t = 100}$ [$\mu M$] | Addition $_{t = 200}$ [$\mu M$] |
|---|---|---|---|
| FBPase$_{ox}$ | 0 | 0 | 0 |
| FBPase$_{red}$ | 4.70756e-3 | 0 | 0 |
| Trxf1$_{ox}$ | 0 | 0 | 0 |
| Trxf1$_{red}$ | 5 | 0 | 5 |
| 2-CysPrx$_{ox}$ | 0 | 0 | 2.5–20 |
| 2-CysPrx$_{red}$ | 0 | 0 | 0 |
| FBP | 0 | 600 | 0 |
| Fru-6-P | 0 | 0 | 0 |

DOI: https://doi.org/10.7554/eLife.38194.027

The enzymatic parameters were either taken from the literature or calculated. *Appendix 1—table 2* summarizes the parameters introduced into the mathematical model.

**Appendix 1—table 2.** Parameters used for modeling the enzyme assay as a reference.

| Parameter | Value | Reference/comment |
|---|---|---|
| $k_1$ | 2.9616e-2 $\mu M^{-1}s^{-1}$ | Calculated from *Collin et al. (2003)* |
| $k_2$ | 1.84e-3 $\mu M^{-1}\ s^{-1}$ | *Collin et al. (2003)* |
| $K_{M(FBP)}$ | 0.028 $\mu M$ | *Pilkis et al., 1987* |
| $k_{cat\_3}$ | 0.1 $s^{-1}$ | *Villadsen and Nielsen, 2001* |
| $Keq_{\_1}$ | 0.5697 | calculated |
| $Keq_{\_2}$ | 0.3856 | calculated |
| FBPase$_{total}$ | 4.70756 $\mu M$ | calculated |
| 2-CysPrx$_{total}$ | 0–20 $\mu M$ | Enzyme test concentration |
| Trxf1$_{total}$ | 5 $\mu M$ | Enzyme test concentration |

DOI: https://doi.org/10.7554/eLife.38194.028

These values and parameters were used to model the relevant enzyme reactions. *Appendix 1—table 3* gives the information on the reactions of Trx-dependent FBPase reduction (Equation 1), the competing reaction of Trx to reduce the oxidized 2-CysPrx (Equation 2) and the enzyme activity of FBPase converting FBP to Fru-6-P (Equation 3). These equations were transformed into enzyme velocity equations describing the reaction rates (*Appendix 1—table 4*).

**Appendix 1—table 3.** Reaction equations describing the model of the enzyme test.

| Reaction number | Reaction |
|---|---|
| 1 | $Trxf1_{red} + FBPase_{ox} A \underset{k_{1-}}{\overset{k_{1+}}{\rightleftharpoons}} BTrxf1_{ox} + FBPase_{red}$ |
| 2 | $Trxf1_{red} + 2 - CysPrx_{ox} A \underset{k_{2-}}{\overset{k_{2+}}{\rightleftharpoons}} BTrxf1_{ox} + 2 - CysPrx_{red}$ |
| 3 | $FBP \overset{FBPase_{red}}{\rightarrow} F6P + P_i$ |

DOI: https://doi.org/10.7554/eLife.38194.029

**Appendix 1—table 4.** Rate expression for the three reactions of the enzyme test model.

| Reaction number | Reaction |
|---|---|
| 1 | $v_1 = k_1 * \left( [Trxf1_{red}] * [FBPase_{ox}] - \frac{[Trxf1_{ox}]*[FBPase_{red}]}{Keq\_1} \right)$ |
| 2 | $v_2 = k_2 * \left( [Trxf1_{red}*[2 - CysPrx_{ox}] - \frac{[Trxf1_{ox}]*[2-CysPrx_{red}]}{K_{eq\_2}} \right)$ |
| 3 | $v_3 = \frac{kcat_3 * [FBPase_{red}]*[FBP]}{[FBP+Km_{FBP}]}$ |

DOI: https://doi.org/10.7554/eLife.38194.030

The rate equations were then implemented using mass action law and Michaelis-Menten kinetics. The equilibrium constants for reaction v1 and v2 were calculated using the redox potentials at pH seven linked to Gibbs free energy (**Poughon et al., 2001**)

$$K_{eq} = e^{\frac{-\Delta G\circ}{RT}}$$

For simplification only FBP was considered in rate reaction v3. The differential equation for the variable read as folows. These variables were calculates as a function of time.

$$\frac{d[Trxf1_{ox}]}{dt} = +v_1 + v_2$$

$$\frac{d[Trxf1_{red}]}{dt} = -v_1 - v_2$$

$$\frac{d[FBPase_{ox}]}{dt} = -v_1$$

$$\frac{d[FBPase_{red}]}{dt} = +v_1$$

$$\frac{d[2-CysPrx_{ox}]}{dt} = -v_2$$

$$\frac{d[2-CysPrx_{red}]}{dt} = +v_2$$

$$\frac{d[FBP]}{dt} = -v_3$$

$$\frac{d[F6P]}{dt} = +v_3$$

## Appendix 2

DOI: https://doi.org/10.7554/eLife.38194.031

### Modeling and simulation of redox changes of FBPase and Trx-f1 upon darkening

This kinetic model assumes light-driven reduction of ferredoxin (Fd) and Fd-dependent thioredoxin reductase during the first 200 s. Then Fd and FTR are switched to full oxidation. Following this simulated light switch-off, the oxidized fractions of Trx and FBPase increased at the expense of reduced fraction. The model was based on the variables as compiled in *Appendix 2—table 1*.

**Appendix 2—table 1.** Values of variables used for modeling the light-dark-transitions.

| component | start value [μM] | Reference/comment |
|---|---|---|
| $FTR_{ox}$ | 0 | Assumption in light 100% reduced |
| $FTR_{red}$ | 4.7727 | calculated from *Yoshida and Hisabori (2017)* |
| $Trxf1_{ox}$ | 3.798e-1 | 20% oxidized |
| $Trxf1_{red}$ | 1.5192 | 80% reduced |
| $FBPase_{ox}$ | 1.426528 | 20% oxidized |
| $FBPase_{red}$ | 5.706112 | 80% reduced |

DOI: https://doi.org/10.7554/eLife.38194.032

Then the minimal model for light-dark-transitions was build consisting of five components; Fd, FTR, Trxf1, FBPase and 2-CysPrx. Fd and 2-CysPrx were clamped to 50% reduced Fd, and 66% oxidized and 34% reduced 2-CysPrx. The physiological concentrations are calculated for 1 mg chlorophyll. *Appendix 2—table 2* gives the parameters taken for the rate constants, equilibrium constants and concentrations for implementing the model. The model was comprised of four reaction equations (*Appendix 2—table 3*) allowing to derive the rate equations given in *Appendix 2—table 4*.

**Appendix 2—table 2.** Parameters used for modeling the light-dark-transitions.

| parameter | value | Reference/comment |
|---|---|---|
| $k_1$ | $7.7047e^{-2}$ $\mu M^{-1}\mu M^{-1}s^{-1}$ | Fitted |
| $k_2$ | $6.819e^{-2}$ $\mu M^{-1}s^{-1}$ | Fitted |
| $k_3$ | $2.9616e^{-2}$ $\mu M^{-1}s^{-1}$ | Calculated from *Collin et al. (2003)* |
| $k_4$ | $1.84e^{-3}$ $\mu M^{-1}s^{-1}$ | *Collin et al. (2003)* |
| $Keq_{Trxf1FBPase}$ | 0.5697 | Calculated |
| $Keq_{Trxf12CP}$ | 0.3856 | Calculated |
| $Fd_{total}$ | 69 μM | calculated from *Peltier et al. (2006)* |
| $Fd_{red\_fix}$ | 34.5 μM | Estimated (50% reduced) |
| $FTR_{total}$ | 4.7727 μM | calculated from *Yoshida and Hisabori (2017)* |
| $Trxf1_{total}$ | 1.899 μM | calculated from *Peltier et al. (2006)* |
| $FBPase_{total}$ | 7.13267 μM | calculated from *Peltier et al. (2006)* |
| $2\text{-}CysPrx_{total}$ | 63.3 μM | Calculated from *Peltier et al. (2006)* |
| $2\text{-}CysPrx_{red\_fix}$ | 21.522 μM | Calculated |
| $2\text{-}CysPrx_{ox\_fix}$ | 41.778 | Calculated |

DOI: https://doi.org/10.7554/eLife.38194.033

**Appendix 2—table 3.** Reaction equation describing the model of the light-dark-transitions.

| reaction number | reaction |
|---|---|
| 1 | $Fd_{red} + Fd_{red} + FTR_{ox} k_1 Fd_{ox} + Fd_{ox} + FTR_{red}$ |
| 2 | $FTR_{red} + Trxf1_{ox} k_2 FTR_{ox} + Trxf1_{red}$ |
| 3 | $Trxf1_{red} + FBPase_{ox} A \underset{k_3^-}{\overset{k_3^+}{\rightleftharpoons}} BTrxf1_{ox} + FBPase_{red}$ |
| 4 | $Trxf1_{red} + 2 - CysPrx_{ox} A \underset{k_2^-}{\overset{k_2^+}{\rightleftharpoons}} BTrxf1_{ox} + 2 - CysPrx_{red}$ |

DOI: https://doi.org/10.7554/eLife.38194.034

**Appendix 2—table 4.** Rate expressions for the four reactions considered for the model of light-dark-transition .

| reaction number | Reaction |
|---|---|
| 1 | $v_1 = k_1 * [Fd_{red}] * [Fd_{red}] * [FTR_{ox}]$ |
| 2 | $v_2 = k_2 * [FTR_{red} * [Trxf1_{ox}]$ |
| 3 | $v_3 = k_3 * \left( [Trxf1_{red}] * [FBPase_{ox}] - \frac{[Trxf1_{ox}] * [FBPase_{red}]}{K_{eq\_Trxf1FBPase}} \right)$ |
| 4 | $v_4 = k_4 * \left( [Trxf1_{red} * [2 - CysPrx_{ox}] - \frac{[Trxf1_{ox}] * [2 - CysPrx_{red}]}{K_{eq\_Trxf12CP}} \right)$ |

DOI: https://doi.org/10.7554/eLife.38194.035

The differential equations of the variables used for simulating light-dark-transitions were constructed as follows:

$$\frac{d[FTR_{ox}]}{dt} = -v_1 + v_2$$

$$\frac{d[FTR_{red}]}{dt} + v_1 - v_2$$

$$\frac{d[Trxf1_{ox}]}{dt} = -v_2 - v_3 - v_4$$

$$\frac{d[Trxf1red]}{dt} = +v_2 - v_3 - v_4$$

$$\frac{d[FBPase_{ox}]}{dt} = -v_3$$

$$\frac{d[FBPase_{red}]}{dt} = +v_3$$

