## [Decision Letter]

[Editors’ note: this article was originally rejected after discussions between the reviewers, but the authors were invited to resubmit after an appeal against the decision.]

Thank you for submitting your work entitled "The chloroplast 2-cysteine peroxiredoxin functions as thioredoxin oxidase in redox regulation of chloroplast metabolism" for consideration by *eLife*. Your article has been reviewed by two peer reviewers, and the evaluation has been overseen by a Reviewing Editor and a Senior Editor. The following individual involved in review of your submission has agreed to reveal their identity: Jeane-Philippe Reichheld (Reviewer #2).

Our decision has been reached after consultation between the reviewers. Based on these discussions and the individual reviews below, we regret to inform you that your work will not be considered further for publication in *eLife*. While one reviewer comments that the experiments are well designed and performed, and the data are clearly presented, a key concern of the other is the general lack of detail throughout the manuscript, particularly regarding the experimental design and the mutants used. This makes understanding and interpretation of the data very difficult.

*Reviewer #1:*

This manuscript concerns a very important question in plant redox biology that has long remained unanswered i.e. how are the thiol-activated enzymes of photosynthesis inactivated at low light or following the transition to darkness. Until now, the mechanisms that achieve efficient oxidation of target proteins were largely unknown although it has long been recognised that oxidised thioredoxins can oxidise the thiol-modulated enzymes of the Calvin-Benson cycle. The data presented in this manuscript shows that 2-CysPrx can also function as a peroxide-dependent thioredoxin oxidase.

The data reported in this manuscript are potentially very interesting. However, insufficient details are provided about the different lines used in the study. The 2cysprxAB and cyclophilin 20-3 mutants are not described in the manuscript and the Materials and methods section does not mention the cyclophilin 20-3 mutants. The Materials and methods section describes experiments done only on the Col-0 wildtype and the 2cysprxAB double insertion line.

In vitro assays using stroma proteins, Trx-f1 and oxidized 2-CysPrx, showed that Trx-f1 can oxidise and inactivate FBPase by transferring electrons from reduced FBPase to 2-CysPrx_ox_. This ability was lost in site-directed mutated variants of 2-CysPrx and also the hyperoxidation mimicking C54D. Hence, the redox activity of both proteins is needed. Based on similar assays, the authors rank the redox proteins in order of inhibition efficiency: Trx-m1>CDSP32>Trx-f1>Trx-x>Trx-m4.

While the results from the in vitro assays generally support the conclusion that the chloroplast 2-CysPrx functions as a peroxide-dependent thioredoxin oxidase, I am concerned that all the analyses were performed only on one mutant line for each gene. Given the problems that can be associated with insertion and other mutants it is now usual to analyse at least two mutants for each gene.

*Reviewer #2:*

The manuscript by Vaseghi et al. describes a thioredoxin oxidase activity of the chloroplastic 2-Cys peroxiredoxin (Prx) and shows that 2-CysPrx serves as an electron sink in the thiol network which is important to oxidize reductively activated proteins. This work also describes a mechanism by which the abundant thioredoxin target 2-CysPrx withdraws electron flux from metabolic enzymes like MDH and PRK and acts to inactivate them under fluctuating light conditions. The work addresses an important and unexplored question of the oxidizing functions of thiol reductases, and is of great interest for the redox field. The manuscript is based on biochemical experiments performed on recombinant proteins, which are supported by genetic data. The experiments are well designed and performed. Data are clearly presented. Main figures are all supported by several supplemental data, which generally make the data solid. The statistical analyses are also well performed.

Figure 5A: The determination of initial NADPH-MDH activity in plant extracts looks tricky. As the delay in the MDH decrease is so transient (<1 min) between 2-CysPrx and WT, how can we be sure that it is not affected during preparation of the protein extracts? How can we explain that the difference is so transient?

To homogenize the data, it would better to represent the leaf MDH (initial/total) timecourse, similarly to Figure 5B for PRK. This would better show that NADPH-MDH and PRK activities no not behave similarly after darkening in 2-CysPrx and WT. This difference should be better discussed.

Subsection “Thermodynamics of oxidation”, first paragraph: The authors suggest that PRK deactivated slowly in WT, while its activity remained unchanged in 2-CysPrx. This is not supported by Figure 5B which suggests that both WT and 2-CysPrx activities decrease rapidly after darkening. How can we explain this decrease?

Figure 7: The authors state that WT and 2-CysPrx plants are growing similarly and that anthocyanin accumulate in WT plants (Results, last paragraph). This is not consistent to the figure. It seems that the WT has much more leaves that the mutant, suggesting that the WT grows more rapidly than 2-CysPrx mutant plants. Counting the rosette leaves in both WT and 2-CysPrx plants at the same stage would help to convince the reader.

Results, last paragraph: How can we explain plant damage in 40/5 s D/H conditions? And why is 2-CysPrx less?

---

## [Author Response]

[Editors’ note: the author responses to the first round of peer review follow.]

Reviewer #1:[…] The data reported in this manuscript are potentially very interesting. However, insufficient details are provided about the different lines used in the study. The 2cysprxAB and cyclophilin 20-3 mutants are not described in the manuscript and the Materials and methods section does not mention the cyclophilin 20-3 mutants. The Materials and methods section describes experiments done only on the Col-0 wildtype and the 2cysprxAB double insertion line.

Thanks for this comment and we apologize for this mistake. We have extended our description of the genetic material we used. Each line was used in other studies before. We now provide the accession numbers and the references.

In vitro assays using stroma proteins, Trx-f1 and oxidized, showed that Trx-f1 can oxidise and inactivate FBPase by transferring electrons from reduced FBPase to ox. This ability was lost in site-directed mutated variants of and also the hyperoxidation mimicking C54D. Hence, the redox activity of both proteins is needed. Based on similar assays, the authors rank the redox proteins in order of inhibition efficiency: Trx-m1>CDSP32>Trx-f1>Trx-x>Trx-m4.

Thanks for this very positive and summarizing statement.

While the results from the in vitro assays generally support the conclusion that the chloroplast functions as a peroxide-dependent thioredoxin oxidase, I am concerned that all the analyses were performed only on one mutant line for each gene. Given the problems that can be associated with insertion and other mutants it is now usual to analyse at least two mutants for each gene.

Thanks for this comment. There are no independent double insertion lines for A and B available. Thus the other double mutant used in Prof. Javier Cejudo’s group is partly the same insertion line. To cope with this important issue, we have included now data of complementation studies as these are usually accepted as proof that off-site effects do not contribute to the observed responses. Thus in the double insertion background of 2cysprxAB we have introduced A under control of the endogenous A promoter. Two independent lines 2cysprxAB/2CysPrxA were selected and confirmed (supporting material) and tested for their response in several experiments: 1) Inactivation of MDH after 10 seconds relative to wildtype and 2cysprxAB, 2) inactivation of PRK after 5 min relative to wildtype and 2cysprxAB, 3) reoxidation kinetics of ferredoxin after illumination of dark adapted leaves and 4) growth in fluctuating light. In each of the cases the complemented lines showed the characteristics of WT proving that the effects seen in *2cysprxAB* indeed were linked to the loss of 2CysPrx and could be reversed by introducing 2CysPrxA.

Reviewer #2:[…] Figure 5A: The determination of initial NADPH-MDH activity in plant extracts looks tricky. As the delay in the MDH decrease is so transient (<1 min) between 2-CysPrx and WT, how can we be sure that it is not affected during preparation of the protein extracts? How can we explain that the difference is so transient?

It is described in the literature that MDH most sensitively responds to changes in stromal redox potentials and this occurs at very negative redox potentials. In addition, metabolite levels strongly contribute to the inactivation and activation. This was just recently competently summarized in the Trends in Plant Science article by Knüsting and Scheibe, 2018. Thus redox-regulation is important, but embedded in a more complex regulatory scenario. We have added data from another experiment. Naturally this type of experiment depends on careful timing and experimental planning, and the statistical analysis gives an indication of reliability. But together with the in vitro experiments, the evidence is quite conclusive, namely that 2-CysPrx contributes to inactivation of MDH.

To homogenize the data, it would better to represent the leaf MDH (initial/total) timecourse, similarly to Figure 5B for PRK. This would better show that NADPH-MDH and PRK activities no not behave similarly after darkening in 2-CysPrx and WT. This difference should be better discussed.

Naturally, we discussed this point in our group before and ask to keep the presentation as is. The reason is that Total NADPH-dependent MDH activity is somewhat higher (about 25%) in 2cysprx prior to darkening. This makes the comparison difficult if we show the ratio of initial to total activity. But as mentioned above we have added data from an additional experiment.

Subsection “Thermodynamics of oxidation”, first paragraph: The authors suggest that PRK deactivated slowly in WT, while its activity remained unchanged in 2-CysPrx. This is not supported by Figure 5B which suggests that both WT and 2-CysPrx activities decrease rapidly after darkening. How can we explain this decrease?

Also in case of PRK, we performed additional experiments and present the mean of the expanded data set. It is quite clear that there is variation in activation state and initially the activity is not much changing. This is in line with the much lower midpoint redox potential of the regulatory thiols in PRK. But the difference after 5 minutes is highly reproducible. Furthermore, the two complemented lines behave like WT (Figure 5—figure supplement 2).

Figure 7: The authors state that WT and 2-CysPrx plants are growing similarly and that anthocyanin accumulate in WT plants (Results, last paragraph). This is not consistent to the figure. It seems that the WT has much more leaves that the mutant, suggesting that the WT grows more rapidly than 2-CysPrx mutant plants. Counting the rosette leaves in both WT and 2-CysPrx plants at the same stage would help to convince the reader.

Apparently our description was not clear enough and we have rephrased it. The 2cysprxAB double mutant grows slower, develops less leaves and the leaf shape is more round than that of WT. The complemented lines behave like wildtype. Only under continuous light, the *2cysprxAB* line develops leaves which resemble WT leaves and the mutant line also grows better as indicated by increased biomass. We find the parameter biomass more important than leaf number in this context.

Results, last paragraph: How can we explain plant damage in 40/5 s D/H conditions? And why is 2-CysPrx less?

This is an interesting question. Naturally we like to link it to the less tight regulation of photosynthesis in 2cysprxAB. Thus possibly, even WT can activate the Calvin cycle slightly during 80 s high light, but fails to do so in 40 s, while the 2cysprxAB mutant even can convert some light energy in photochemistry during the 40 s period. This would mean that oxidative load and thus possibly damage is higher in WT than in 2cysprxAB.